# Towards Enabling Meta-Learning
# from Target Models*

**Su Lu**      **Han-Jia Ye**      **Le Gan**      **De-Chuan Zhan**
State Key Laboratory for Novel Software Technology
Nanjing University, Nanjing, 210023, China
{lus,yehj}@lamda.nju.edu.cn, {ganl,zhandc}@nju.edu.cn

## Abstract

Meta-learning can extract an inductive bias from previous learning experience and assist the training of new tasks. It is often realized through optimizing a meta-model with the evaluation loss of task-specific solvers. Most existing algorithms sample non-overlapping *support* sets and *query* sets to train and evaluate the solvers respectively due to simplicity ($\mathcal{S}/\mathcal{Q}$ protocol). Different from $\mathcal{S}/\mathcal{Q}$ protocol, we can also evaluate a task-specific solver by comparing it to a target model $\mathcal{T}$, which is the optimal model for this task or a model that behaves well enough on this task ($\mathcal{S}/\mathcal{T}$ protocol). Although being short of research, $\mathcal{S}/\mathcal{T}$ protocol has unique advantages such as offering more informative supervision, but it is computationally expensive. This paper looks into this special evaluation method and takes a step towards putting it into practice. We find that with a small ratio of tasks armed with target models, classic meta-learning algorithms can be improved a lot without consuming many resources. We empirically verify the effectiveness of $\mathcal{S}/\mathcal{T}$ protocol in a typical application of meta-learning, *i.e.*, few-shot learning. In detail, after constructing target models by fine-tuning the pre-trained network on those hard tasks, we match the task-specific solvers and target models via knowledge distillation.

## 1   Introduction

Meta-learning means improving performance measures over a family of tasks by their training experience [22]. It has been researched in various fields such as image classification [11, 16] and reinforcement learning [6, 14]. By reusing transferable meta-knowledge extracted from previous tasks, we can learn new tasks with a higher efficiency or a shortage of data.

A typical meta-learning algorithm can be decomposed into two iterative phases. In the first phase, we train a solver of a task on its training set with assistance of meta-model. In the second phase, we optimize the solver's performance to update meta-model. One key factor in this procedure is the way to evaluate the solver because the evaluation result acts as the supervision signal for meta-model. Early meta-learning algorithms [19, 23] directly use the solver's training loss as its performance metric, and optimize this metric over a distribution of tasks. Obviously, inner-task over-fitting may happen during the training of task-specific solvers, resulting in an inaccurate supervision signal for the meta-model. This drawback is even more amplified in applications where the training set of each task is limited such as few-shot learning and noisy learning.

Intuitively, assessment of solvers should be independent of their training sets. This principle draws forth two important meta-learning algorithms in 2016 [24, 28], which respectively export solver evaluation from the perspective of "data" and "model". In this paper, we call these two methodologies $\mathcal{S}/\mathcal{Q}$ protocol and $\mathcal{S}/\mathcal{T}$ protocol. In $\mathcal{S}/\mathcal{Q}$ protocol, $\mathcal{S}$ means *support* set and $\mathcal{Q}$ means *query* set. They

---

*De-Chuan Zhan is the corresponding author.

35th Conference on Neural Information Processing Systems (NeurIPS 2021).

contain non-overlapping instances sampled from a same distribution. By training the solver on $\mathcal{S}$ and evaluating it on $\mathcal{Q}$, we are able to obtain an approximate generalization error of the solver and eventually provide the meta-model with a reliable supervision signal. Another choice is to compare the trained solver with an ideal target model $\mathcal{T}$. Assuming that $\mathcal{T}$ works well on a task, we can minimize the discrepancy between the trained solver and $\mathcal{T}$ to pull the solver closer to $\mathcal{T}$. Here $\mathcal{T}$ can be Bayesian optimal solution to a task or a model trained on a sufficiently informative dataset. Figure 1 gives an illustration of both $\mathcal{S}/\mathcal{Q}$ protocol and $\mathcal{S}/\mathcal{T}$ protocol.

Although appeared in the same year, $\mathcal{S}/\mathcal{Q}$ protocol is more widely accepted by meta-learning society [4, 8, 13, 10] while the research about how to leverage target models remains immature. The main reason is the simplicity of $\mathcal{S}/\mathcal{Q}$ and the computational hardness of $\mathcal{S}/\mathcal{T}$. However, $\mathcal{S}/\mathcal{T}$ protocol has some unique advantages. Firstly, it does not depend on possibly biased and noisy *query* sets. Secondly, by viewing *support* sets and their corresponding target models as (feature, label) samples, meta-learning is reduced to supervised learning and we can transfer insights from supervised learning to improve meta-learning [2]. Thirdly, we can treat the target model as a teacher and incorporate a teacher-student framework like knowledge distillation [7] and curriculum learning [1] in meta-learning. Thus, it is necessary and meaningful to study $\mathcal{S}/\mathcal{T}$ protocol in meta-learning.

This paper looks into $\mathcal{S}/\mathcal{T}$ protocol and takes a step towards enabling meta-learning from target models. We mainly answer two questions: (1) If we already have access to target models, how to learn from them? What are the benefits of learning from them? (2) In a real-world application, how to obtain target models efficiently and make $\mathcal{S}/\mathcal{T}$ protocol computationally tractable? For the first question, we propose to match the

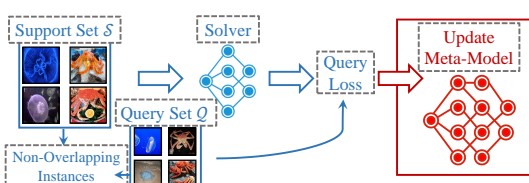

(a) $\mathcal{S}/\mathcal{Q}$ protocol for meta-learning.

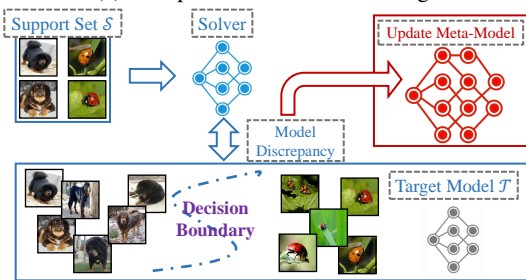

(b) $\mathcal{S}/\mathcal{T}$ protocol for meta-learning.

Figure 1: Comparison between $\mathcal{S}/\mathcal{Q}$ protocol and $\mathcal{S}/\mathcal{T}$ protocol. (a) In $\mathcal{S}/\mathcal{Q}$ protocol, each task contains a *support* set $\mathcal{S}$ and a *query* set $\mathcal{Q}$. We train a solver on $\mathcal{S}$ and evaluate it on $\mathcal{Q}$, and *query* loss is used to optimize meta-model. (b) In $\mathcal{S}/\mathcal{T}$ protocol, each task contains a *support* set $\mathcal{S}$ and a target model $\mathcal{T}$. After training a solver on $\mathcal{S}$, we directly minimize the discrepancy between it and $\mathcal{T}$.

task-specific solver to the target model in output space. Learning from target models brings us more robust solvers. For the second question, we focus on a typical application scenario of meta-learning, *i.e.*, few-shot learning. We construct target models by fine-tuning the globally pre-trained network on those hard tasks to maintain efficiency.

## 2   Related Work

**Meta-Learning.**   Meta-learning aims at extracting task-level experience (so-called meta-knowledge) from seen tasks, while generalizing the learned meta-knowledge to unseen tasks efficiently. Researchers have studied several kinds of meta-knowledge like model initialization [3, 25], embedding network [21, 12, 9, 20, 4], external memory [19, 5], optimization strategy [15, 18], and data augmentation strategy [13]. Despite their diversity in meta-knowledge, most existing models are trained under $\mathcal{S}/\mathcal{Q}$ protocol, and rely on a randomly sampled and possibly biased *query* set. Actually, most algorithms are protocol-agnostic, and both $\mathcal{S}/\mathcal{Q}$ protocol and $\mathcal{S}/\mathcal{T}$ protocol can be applied to them. Thus, our work on $\mathcal{S}/\mathcal{T}$ is general, and it has a wide application field.

**Learning from Target Models.**   The idea of learning from target models in meta-learning is first proposed by [28]. In [28], the authors constructed a model regression network that explicitly regresses between small-sample classifiers and target models in parameter space. Here both solvers and target models are limited to low-dimensional linear classifier, making it feasible to regress between them. From our perspective, matching two models' parameters is not practical when the dimension of parameters is too high. Thus, we match two models in output space in this paper. Similarly, there are

other papers focusing on meta-learning from target models [27, 31]. The most similar work to us is [31], which constructs target models with abundant instances and matches task-specific solvers and target models. However, they all assume that every single task has a target model, increasing both space and time complexity of $\mathcal{S}/\mathcal{T}$ protocol. To summarize, we claim that one key point in putting $\mathcal{S}/\mathcal{T}$ protocol into practice is reducing the requirement for target models. In this paper, we focus on those hard tasks, and find that by learning from a small ratio of informative target models, classic meta-learning algorithms can be improved.

## 3 Preliminary

Meta-learning extracts high-level knowledge by a meta-model from *meta-training* tasks sampled from a task distribution $p(\boldsymbol{\tau})$ and reuses the learned meta-model on new tasks belonging to the same distribution. Each task $\boldsymbol{\tau}$ has a task-specific *support* set $\mathcal{S} = \{(\mathbf{x}_i, \mathbf{y}_i)\}_{i=1}^{|\mathcal{S}|}$, and we can train on $\mathcal{S}$ a solver $g : \mathbb{X} \to \mathbb{Y}$ parameterized by $\boldsymbol{\gamma}_g$. Without loss of generality, a meta-model can be defined as $f : \mathbb{S} \to \mathbb{G}$ parameterized by $\boldsymbol{\theta}_f$ that receives a *support* set as input and outputs a solver. Here $\mathbb{S}$ is the space of *support* sets and $\mathbb{G}$ is the space of solvers. In other words, $f$ encodes the training process of $g$ on $\mathcal{S}$ under the supervision of meta-knowledge $\boldsymbol{\theta}_f$. Taking two well-known meta-learning algorithms, MAML [3] and ProtoNet [21], as examples, we have the following concrete forms of $f$:

- MAML meta-learns a model initialization $\boldsymbol{\theta}_f$ and fine-tunes it on each $\mathcal{S}$ with one gradient descent step to obtain a task-specific solver $g$. It can be written as Equ (1). $\eta$ is step size and $\ell : \mathbb{Y} \times \mathbb{Y} \to \mathbb{R}_+$ is some loss function.

$$f(\mathcal{S}; \boldsymbol{\theta}_f) = g\left(\,\cdot\,; \boldsymbol{\theta}_f - \eta \, \nabla_{\boldsymbol{\gamma}} \sum_{(\mathbf{x}_i, \mathbf{y}_i) \in \mathcal{S}} \ell\left(g(\mathbf{x}_i; \boldsymbol{\gamma}), \mathbf{y}_i\right)\bigg|_{\boldsymbol{\gamma} = \boldsymbol{\theta}_f}\right) \tag{1}$$

- ProtoNet meta-learns an embedding function $\phi_{\boldsymbol{\theta}_f}$ parameterized by $\boldsymbol{\theta}_f$ and generates a lazy solver which classifies an instance to the category of its nearest class center. Here $g$ is implicitly parameterized by both $\boldsymbol{\theta}_f$ and embedded *support* instances.

$$f(\mathcal{S}; \boldsymbol{\theta}_f) = g\left(\,\cdot\,; \boldsymbol{\theta}_f, \{\phi_{\boldsymbol{\theta}_f}(\mathbf{x}_i) | (\mathbf{x}_i, \mathbf{y}_i) \in \mathcal{S}\}\right) \tag{2}$$

$\mathcal{S}/\mathcal{Q}$ **Protocol.** How to evaluate the solver $g$ trained on $\mathcal{S}$? The answer to this question differs conventional $\mathcal{S}/\mathcal{Q}$ protocol [24] from $\mathcal{S}/\mathcal{T}$ protocol. In $\mathcal{S}/\mathcal{Q}$ protocol, we sample another *query* set $\mathcal{Q} = \{\mathbf{x}_j, \mathbf{y}_j\}_{j=1}^{|\mathcal{Q}|}$ apart from $\mathcal{S}$ for each task. Instances in $\mathcal{S}$ and $\mathcal{Q}$ are *i.i.d.* distributed and have a same label set, and we evaluate $g$ by its loss on $\mathcal{Q}$. Since $\mathcal{S}$ and $\mathcal{Q}$ contain non-overlapping instances, loss on $\mathcal{Q}$ is a more reliable supervision signal. $\mathcal{S}/\mathcal{Q}$ protocol can be formulated as Equ (3). Here $\mathcal{D}^{\mathrm{tr}}$ is the *meta-training* set and we can sample *meta-training* tasks $\boldsymbol{\tau}^{\mathrm{tr}}$ from it.

$$\min_f \sum_{\boldsymbol{\tau}^{\mathrm{tr}} = (\mathcal{S}^{\mathrm{tr}}, \mathcal{Q}^{\mathrm{tr}}) \in \mathcal{D}^{\mathrm{tr}}} \sum_{(\mathbf{x}_j, \mathbf{y}_j) \in \mathcal{Q}^{\mathrm{tr}}} \ell(f(\mathcal{S}^{\mathrm{tr}})(\mathbf{x}_j), \mathbf{y}_j) \tag{3}$$

$\mathcal{S}/\mathcal{T}$ **Protocol.** Any sampled *query* set $\mathcal{Q}$ can be biased and noisy, which may cause an inaccurate evaluation of the solver. An alternative is directly matching the task-specific solver $g = f(\mathcal{S})$ and a target model $\mathcal{T}$ that works well on the corresponding task. By computing the distance from the solver to target model, we obtain a more robust training signal to update meta-model. By replacing the solver evaluation part in Equ (3), we have the following $\mathcal{S}/\mathcal{T}$ protocol Equ (4). Here $\mathcal{L} : \mathbb{G} \times \mathbb{G} \to \mathbb{R}_+$ is some loss function to measure the discrepancy between $g$ and target model $\mathcal{T}$.

$$\min_f \sum_{\boldsymbol{\tau}^{\mathrm{tr}} = (\mathcal{S}^{\mathrm{tr}}, \mathcal{T}^{\mathrm{tr}}) \in \mathcal{D}^{\mathrm{tr}}} \mathcal{L}(f(\mathcal{S}^{\mathrm{tr}}), \mathcal{T}^{\mathrm{tr}}) \tag{4}$$

## 4 Effect of Target Model

We have introduced some basic concepts in meta-learning, and formulate $\mathcal{S}/\mathcal{Q}$ protocol and $\mathcal{S}/\mathcal{T}$ protocol in Section 3. In this section, we assume that target models are available, and study how

to utilize them to assist meta-learning. Firstly, we propose a model matching framework based on output comparison. Secondly, we verify the effectiveness of our proposal in a synthetic experiment. Moreover, we try to decrease the ratio of tasks that have target models, and show that it is possible to reduce the resource consumption of $\mathcal{S}/\mathcal{T}$ protocol.

## 4.1 Model Matching

In $\mathcal{S}/\mathcal{T}$ protocol, one key point is how to match the solver $g$ and its target model $\mathcal{T}$. In other words, we need to specify the concrete formulation of $\mathcal{L}(g, \mathcal{T})$. Generally, methods to match $g$ to $\mathcal{T}$ can be classified into two categories. Firstly, we can directly match two models' parameters or use another model to regress between two models' parameters [28]. For example, let $\gamma_g$ and $\gamma_{\mathcal{T}}$ be the parameters of $g$ and $\mathcal{T}$, we can set $\mathcal{L}(g, \mathcal{T}) = \sum_{(\mathbf{x}_i, \mathbf{y}_i) \in \mathcal{S}^{\text{tr}}} \ell(g(\mathbf{x}_i), \mathbf{y}_i) + \lambda \|\gamma_g - \gamma_{\mathcal{T}}\|_2^2$. Here $\lambda$ is a balancing hyperparameter. This method may work well for

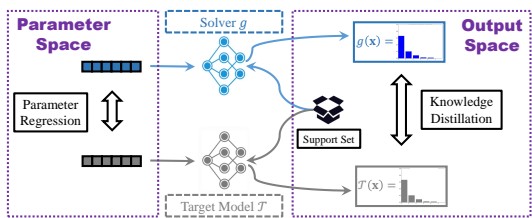

Figure 2: Two approaches to matching $g$ and target model $\mathcal{T}$. Left: matching them in parameter space. Right: matching them in output space.

low-dimensional parameters, but is not suitable for complex models like deep neural networks. A better alternative is to match two models in their output space, *i.e.*, $\mathcal{L}(g, \mathcal{T}) = \sum_{(\mathbf{x}_i, \mathbf{y}_i) \in \mathcal{S}^{\text{tr}}} [(1 - \lambda)\ell(g(\mathbf{x}_i), \mathbf{y}_i) + \lambda \mathbf{D}(\mathcal{T}(\mathbf{x}_i), g(\mathbf{x}_i))]$. Here $\mathbf{D}(\cdot, \cdot)$ is a function that measures the discrepancy between $\mathcal{T}(\mathbf{x}_i)$ and $g(\mathbf{x}_i)$. If we instantiate $\mathbf{D}(\cdot, \cdot)$ as KL divergence $\mathbf{KL}(\cdot\|\cdot)$ for classification problem, the aforementioned loss function is equivalent to that of knowledge distillation. Figure 2 is an illustration of approaches to matching a solver to a target model.

## 4.2 Empirical Study: Sinusoid Regression

In this part, we assume that target models are available, and evaluate the effectiveness of our proposed matching approach. We construct a synthetic regression problem, and try to answer the following questions: (1) Can $\mathcal{S}/\mathcal{T}$ protocol outperform $\mathcal{S}/\mathcal{Q}$ protocol when target models are available? (2) Is it possible to improve meta-learning with only a few target models?

**Setting.** Consider regression tasks $\mathcal{T}(x) = a \sin(bx - c)$ where $a$, $b$, and $c$ are uniformly sampled from $[0.1, 5]$, $[0.5, 2]$, and $[0.5, 2\pi]$ respectively. For each task, we generate 10 *support* instances by uniformly sampling $x$ in range $[-5, 5]$. For $\mathcal{S}/\mathcal{Q}$ protocol, we additionally sample 30 *query* instances for each task. We then set $y = \mathcal{T}(x) + \epsilon$ where $\epsilon \sim \mathcal{N}(0, 0.5)$ is a Gaussian noise. 10000 tasks are used for both *meta-training* and *meta-testing*. 500 tasks are used for *meta-validation*.

**Algorithms.** We consider two classic meta-learning algorithms, MAML [3] and ProtoNet [21]. MAML can be directly applied to a regression task, but ProtoNet is originally designed for classification. In this part, we modify ProtoNet slightly to fit regression problem. In detail, we try to meta-learn an embedding function $\phi : \mathbb{R} \to \mathbb{R}^{100}$, with assistance of which the similarity-based regression model $g(\cdot ; \{\phi(x_i) | (x_i, y_i) \in \mathcal{S}\})$ works well across all tasks. Here for any instance $(x, y)$, $\hat{y} = g(x) = \sum_{(x_i, y_i) \in \mathcal{S}} w_i y_i$ and $w_i = \frac{\exp\{\langle \phi(x_i), \phi(x) \rangle\}}{\sum \exp\{\langle \phi(x_i), \phi(x) \rangle\}}$. A same embedding network is used in two algorithms. We train MAML and ProtoNet under $\mathcal{S}/\mathcal{Q}$ protocol and $\mathcal{S}/\mathcal{T}$ protocol. When using $\mathcal{S}/\mathcal{Q}$ protocol, we minimize MSE loss on 30 *query* instances to optimize $\phi$. For $\mathcal{S}/\mathcal{T}$ protocol, we match the solver and the target model in output space, and set $\mathbf{D}(\mathcal{T}(x_i), g(x_i)) = \|\mathcal{T}(x_i) - g(x_i)\|_2^2$. Thus, the loss function under $\mathcal{S}/\mathcal{T}$ protocol is $\mathcal{L}(g, \mathcal{T}) = \sum_{(x_i, y_i) \in \mathcal{S}^{\text{tr}}} [(1 - \lambda)\|g(x_i) - y_i\|_2^2 + \lambda\|g(x_i) - \mathcal{T}(x_i)\|_2^2]$. $\lambda$ is a hyper-parameter. More implementation details can be found in the supplementary material.

**Superiority of $\mathcal{S}/\mathcal{T}$ Protocol.** Table 1 shows the MSE of four models on *meta-testing* tasks. We can see that models trained under $\mathcal{S}/\mathcal{T}$ protocol consistently outperform models trained under $\mathcal{S}/\mathcal{Q}$ protocol. In Figure 3, we visualize a randomly chosen *meta-testing* task. Different colors are used for different meta-learning algorithms, and dotted lines and dashed lines are used for $\mathcal{S}/\mathcal{Q}$ protocol and $\mathcal{S}/\mathcal{T}$ protocol respectively. We can see that models trained under $\mathcal{S}/\mathcal{T}$ protocol fit the target sinusoid

curve better. It is meaningful to discuss why target models improve meta-learning algorithms. In this empirical study, distillation from target models can be interpreted as label denoising. In detail, we can prove[2] that meta-learning loss under $\mathcal{S}/\mathcal{T}$ protocol $(1-\lambda)\|g(x)-y\|_2^2 + \lambda\|g(x)-\mathcal{T}(x)\|_2^2$ is an upper bound of $\|g(x)-(y-\lambda\epsilon)\|_2^2$, which is the standard MSE loss between the output of solver $g$ and cleaner label $y-\lambda\epsilon$ (raw label $y$ equals to $\mathcal{T}(x)+\epsilon$). Therefore, the larger $\lambda$ is, the cleaner training labels are. Table 2 is an ablation study on hyper-parameter $\lambda$. As expected, both algorithms trained under $\mathcal{S}/\mathcal{T}$ achieve better performance with larger $\lambda$. These results demonstrate the superiority of $\mathcal{S}/\mathcal{T}$ protocol when target models are available.

**Reducing the Requirement for Target Models.** Despite the satisfying results in the empirical study, it does not mean that we can apply $\mathcal{S}/\mathcal{T}$ protocol in real-world applications and necessarily obtain higher performance. Up till now, we have assumed that every single *meta-training* task has a target model. This assumption is too strong from two aspects. Firstly, we usually don't have ready-made target models, and constructing target models is not trivial. Secondly, even though we have designed a method to construct target models, it will cost too much time to construct a target model for every single *meta-training* task. Existing researches that focus on meta-learning from target models often bypass this dilemma by restricting the complexity of solvers and target models [28] or building one global target model. In this paper, we study a more general methodology - reducing the number of required target models. If we randomly choose a small subset of *meta-training* tasks, and only provide these tasks' target models, how will the model performance change? To answer the question, we first randomly sample subsets of tasks that have target models, and abandon target models for other tasks. In this case, the meta-learning loss of tasks without target models degenerates to $\mathcal{S}/\mathcal{Q}$ loss. By ranging the size of this subset, we can plot the performance curve of MAML and ProtoNet in Figure 4. Then, we heuristically select the hardest tasks from all *meta-training* tasks and only deploy target models for these tasks. In this regression problem, a sinusoid curve is defined as $a\sin(bx-c)$, and larger $a$ or smaller $b$ induce steeper curves. We simply consider these steep curves as hard tasks, and sort the hardness of all *meta-training* tasks according to $a-b$. Another two performance curves using this heuristic are also plotted in Figure 4. We can see that when using this naive heuristic, we can obtain an evident performance gain with only $500(5\%)$ target models. This finding inspires us to analyse the hardness of tasks in meta-learning, and confirms the possibility of learning from a few target models.

Table 1: Average test MSE of two meta-learning algorithms. Models trained under $\mathcal{S}/\mathcal{T}$ protocol outperform those trained under $\mathcal{S}/\mathcal{Q}$ protocol.

| Method | MAML | | ProtoNet | |
|---|---|---|---|---|
| | $\mathcal{S}/\mathcal{Q}$ | $\mathcal{S}/\mathcal{T}$ | $\mathcal{S}/\mathcal{Q}$ | $\mathcal{S}/\mathcal{T}$ |
| MSE on $\mathcal{D}^{ts}$ | 4.933 | **3.621** | 4.706 | **3.332** |

Table 2: Average test MSE of two meta-learning algorithms with different $\lambda$ values. Larger $\lambda$ offers cleaner labels, resulting in better models.

| $\lambda$ | 1 | 0.8 | 0.5 | 0.2 |
|---|---|---|---|---|
| MSE: MAML($\mathcal{S}/\mathcal{T}$) | **3.220** | 3.419 | 3.621 | 3.833 |
| MSE: ProtoNet($\mathcal{S}/\mathcal{T}$) | **3.137** | 3.304 | 3.332 | 3.550 |

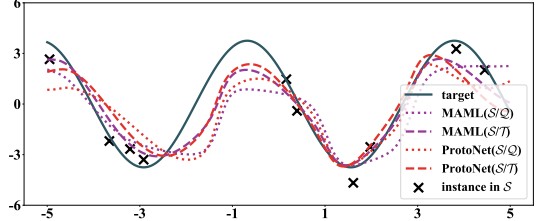

Figure 3: Visualization of a randomly sampled *meta-testing* task. Dotted lines are used for $\mathcal{S}/\mathcal{Q}$ protocol while dashed lines are used for $\mathcal{S}/\mathcal{T}$ protocol. We can see that models trained under $\mathcal{S}/\mathcal{T}$ protocol can fit the target sinusoid curve better.

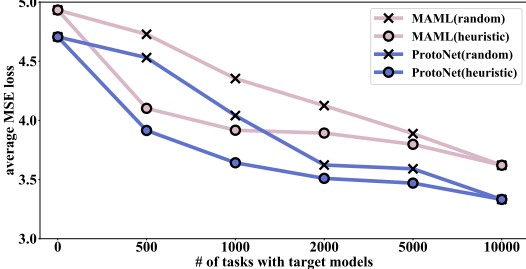

Figure 4: Change of MSE loss over number of *meta-training* tasks that have target models. By selecting hard tasks heuristically, we are able to obtain an evident performance gain with a small number of target models.

---

[2]We leave the proof to the supplementary material.

# 5 Application Case: Few-Shot Learning

Few-shot learning is a typical application of meta-learning. It aims at recognizing new categories with only a few labelled instances. In few-shot learning, we have two datasets that contain non-overlapping classes, *i.e.*, $\mathcal{D}^{\text{tr}}$ and $\mathcal{D}^{\text{ts}}$. $\mathcal{D}^{\text{tr}}$ is composed of seen classes while $\mathcal{D}^{\text{ts}}$ contains unseen classes. We can sample $N$-way $K$-shot[3] *meta-training* tasks from $\mathcal{D}^{\text{tr}}$ to train the meta-model, and expect that the trained meta-model will also work well on $\mathcal{D}^{\text{ts}}$.

## 5.1 Task Hardness

Following the idea of constructing target models for hard tasks, we firstly investigate which tasks are hard in few-shot learning. We consider the relationship between classes as a key factor that determines the hardness of a classification task. Assuming that there are $C^{\text{tr}}$ classes in $\mathcal{D}^{\text{tr}}$, we first compute a similarity matrix $\mathbf{F} \in \mathbb{R}^{C^{\text{tr}} \times C^{\text{tr}}}$ whose element $\mathbf{F}_{uv}$ equals to the similarity between the $u$-th class centre and the $v$-th class center. In few-shot learning, pre-training the backbone network on $\mathcal{D}^{\text{tr}}$ has become a common practice [26, 30], and we can compute these class centres based on the pre-trained model $\phi^{\text{pt}}$ as Equ (5) and Equ (6). In Equ (5), $K_u$ is the number of instances of class $u$ in $\mathcal{D}^{\text{tr}}$, and with a bit abuse of notation, we use $\mathbf{y}_i = u$ to select instances belonging to the $u$-th class.

Table 3: Average accuracy on auxiliary dataset $\mathcal{D}^{\text{au}}$. Fine-tuned target models outperform a single pre-trained target model on randomly sampled tasks.

| Target Model | pre-train | fine-tune |
|---|---|---|
| Accuracy on $\mathcal{D}^{\text{au}}$ | 98.24 | **99.37** |

$$\mathbf{c}_u = \frac{1}{K_u} \sum_{(\mathbf{x}_i, \mathbf{y}_i) \in \mathcal{D}^{\text{tr}} \wedge \mathbf{y}_i = u} \phi^{\text{pt}}(\mathbf{x}_i), \quad u \in [C^{\text{tr}}] \tag{5}$$

$$\mathbf{F}_{uv} = \frac{\mathbf{c}_u \cdot \mathbf{c}_v}{\|\mathbf{c}_u\| \cdot \|\mathbf{c}_v\|}, \quad u, v \in [C^{\text{tr}}] \tag{6}$$

Figure 5: Grouping of 1000 tasks according to their hardness. Both $\phi^{\text{pt}}$ and $\phi^{\text{ft}}$ achieve lower accuracy on harder tasks, verifying the reasonability of our proposed hardness metric. Fine-tuned target models obtain a remarkable performance gain on hard tasks.

With similarity matrix $\mathbf{F}$, we can take out the sub-similarity matrix of task $\boldsymbol{\tau}$ by slicing the rows and columns corresponding to classes contained in $\boldsymbol{\tau}$. The hardness of task $\boldsymbol{\tau}$ is defined as the sum of its sub-similarity matrix. The more similar classes in $\boldsymbol{\tau}$ are, the more difficult to differ them from each other. The hardness of every *meta-training* task can be evaluated with similarity matrix $\mathbf{F}$, and we compute $\mathbf{F}$ only once.

## 5.2 Target Model Construction

As mentioned in last part, pre-training the backbone network on seen classes is a widely used technology in few-shot learning. The pre-trained network $\phi^{\text{pt}}$ is optimized using cross-entropy loss on the whole *meta-training* set, and can classify all classes in $\mathcal{D}^{\text{tr}}$. Since there are $C^{\text{tr}}$ classes in $\mathcal{D}^{\text{tr}}$, the output of $\phi^{\text{tr}}$ is a $C^{\text{tr}}$-dimensional vector. Given a specific $N$-way task $\boldsymbol{\tau}$, a naive approach to obtain a target model is taking out $N$ corresponding dimensions of the pre-trained model's output. However, using a single pre-trained model to assist the meta-learning of all tasks is sub-optimal. We claim that fine-tuning the pre-trained model on the subset of $\mathcal{D}^{\text{tr}}$ that contains classes in $\boldsymbol{\tau}$ can give us a better target model for $\boldsymbol{\tau}$.

**Evaluation on Auxiliary Dataset.** To verify the reasonability of the heuristic task hardness metric and the effectiveness of the fine-tuning approach, we need another auxiliary dataset $\mathcal{D}^{\text{au}}$. $\mathcal{D}^{\text{au}}$ contains same classes as $\mathcal{D}^{\text{tr}}$, and we can evaluate the accuracy of constructed target models (trained on $\mathcal{D}^{\text{tr}}$) on $\mathcal{D}^{\text{au}}$. We conduct an experiment on *mini*ImageNet [24] to check whether fine-tuned target models

---

[3]An $N$-way $K$-shot task is a classification task with $N$ classes and $K$ instances in each class.

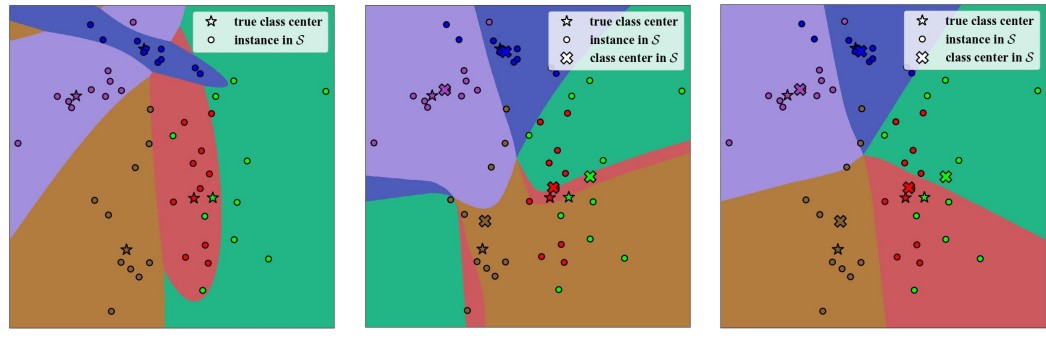

(a) Bayesian optimal classifier.    (b) ProtoNet trained under $\mathcal{S}/\mathcal{Q}$.    (c) ProtoNet trained under $\mathcal{S}/\mathcal{T}$.

Figure 6: Decision boundaries of three different models in raw 2-d space. Different point colors represent different classes, and different background colors represent different classification regions. $5\%$ tasks have target models. A 5-way 10-shot task is visualized. (a) Bayesian optimal model constructed with parameters $\{\boldsymbol{\mu}_n\}_{n=1}^N$ and $\{\boldsymbol{\Sigma}_n\}_{n=1}^N$. Although having the lowest misclassification error in expectation, it is not robust to noises since the decision boundary is very steep. (b) ProtoNet trained under $\mathcal{S}/\mathcal{Q}$ protocol. Decision boundary is still not regular. (c) ProtoNet trained under $\mathcal{S}/\mathcal{T}$ protocol. Decision boundary is very smooth due to the regularization effect of knowledge distillation. Models trained under $\mathcal{S}/\mathcal{T}$ protocol are more robust to noisy of biased instances.

are better than pre-trained target models. Firstly, we pre-train a ResNet-12 with a linear layer on the *meta-training* split of *mini*ImageNet. After that, we randomly sample 1000 5-way tasks from $\mathcal{D}^{\text{tr}}$, and fine-tune the pre-trained backbone to obtain 1000 target models. For each task $\boldsymbol{\tau}$, we take out all instances in $\mathcal{D}^{\text{au}}$ that belong to classes in $\boldsymbol{\tau}$ to evaluate $\phi^{\text{pt}}$ and $\phi_{\boldsymbol{\tau}}^{\text{ft}}$. Table 3 shows the average accuracy on auxiliary dataset $\mathcal{D}^{\text{au}}$. We can see that fine-tuned target models achieve higher accuracy because they are task-specific, but the performance gain is marginal. The pre-trained model already works well enough on these seen classes. This means it is not cost-effective to fine-tune a target model for every single *meta-training* task. In Figure 5, we divide these 1000 tasks into 10 bins according to their hardness. In each bin, we compute the average accuracy of $\phi^{\text{pt}}$ and $\phi^{\text{ft}}$. Now we can draw two conclusions. Firstly, both $\phi^{\text{pt}}$ and $\phi^{\text{ft}}$ achieve lower accuracy on harder tasks, and this verifies the reasonability of our proposed hardness metric. Secondly, the performance gain of fine-tuned target models are most remarkable on hard tasks, and this means fine-tuning target models for hard tasks can simultaneously save computing resources and improve the pre-trained target model.

Now we can summarize our $\mathcal{S}/\mathcal{T}$ protocol for few-shot learning. Firstly, we pre-train $\phi^{\text{pt}}$ on $\mathcal{D}^{\text{tr}}$, and then sample *meta-training* tasks from seen classes. Secondly, we sort the *meta-training* tasks according to their hardness, and fine-tune the pre-trained network to obtain local target models for a small ratio of hardest tasks. Denote by $\mathcal{D}_1^{\text{tr}}$ the set of tasks that have target models and $\mathcal{D}_2^{\text{tr}}$ the set of tasks that do not have target models. For tasks in $\mathcal{D}_1^{\text{tr}}$, we train task-specific solvers on their *support* sets, and then evaluate these solvers under $\mathcal{S}/\mathcal{T}$ protocol. For tasks in $\mathcal{D}_2^{\text{tr}}$, we simply use $\mathcal{S}/\mathcal{Q}$ to compute *query* loss, as shown in Equ (7).

$$
\begin{aligned}
\min_f \sum_{(\mathcal{S}_1^{\text{tr}},\mathcal{T}_1^{\text{tr}})\in\mathcal{D}_1^{\text{tr}}} \sum_{(\mathbf{x}_i,\mathbf{y}_i)\in\mathcal{S}_1^{\text{tr}}} &\left[(1-\lambda)\ell(f(\mathcal{S}_1^{\text{tr}})(\mathbf{x}_i),\mathbf{y}_i)+\lambda\mathbf{KL}(\mathcal{T}_1^{\text{tr}}(\mathbf{x}_i)||f(\mathcal{S}_1^{\text{tr}})(\mathbf{x}_i))\right] \\
&+\sum_{(\mathcal{S}_2^{\text{tr}},\mathcal{Q}_2^{\text{tr}})\in\mathcal{D}_2^{\text{tr}}} \sum_{(\mathbf{x}_j,\mathbf{y}_j)\in\mathcal{Q}_2^{\text{tr}}} \ell(f(\mathcal{S}_2^{\text{tr}})(\mathbf{x}_j),\mathbf{y}_j)
\end{aligned}
\tag{7}
$$

Different from $\mathcal{S}/\mathcal{Q}$ protocol, $\mathcal{S}/\mathcal{T}$ protocol does not rely on randomly sampled *query* sets, and target models usually offer more information than instances. Distillation term plays the role of regularization, enforcing the solvers for hard tasks to be smooth (see next subsection). Although the idea of $\mathcal{S}/\mathcal{T}$ protocol is proposed in 2016, it is not widely used due to its computational intractability. However, in this paper we propose an efficient method to construct target models, and only deploy target models for a small ratio of hard tasks. This opens the door for future research of $\mathcal{S}/\mathcal{T}$ protocol, and unearth the potential of existing meta-learning algorithms.

## 5.3 Empirical Study: Gaussian Classification

In this part, we test our proposed method on a synthetic classification dataset. The purposes of this empirical study are two-fold: (1) check whether $\mathcal{S}/\mathcal{T}$ protocol with only a few target models can improve classic meta-learning algorithm; (2) study why distillation from target models can help.

**Setting.** In this experiment, we randomly generate 100 2-d Gaussian distributions. There are 64 classes for *meta-training*, 16 classes for *meta-validation*, and 20 classes for *meta-testing*. We sample 100 instances for each class to form the whole dataset. For each class, we sample its mean vector $\boldsymbol{\mu} \sim \mathcal{U}^2[-10, 10] \in \mathbb{R}^2$ and covariance matrix $\boldsymbol{\Sigma} = \boldsymbol{\Sigma}'^\top \boldsymbol{\Sigma}'$ where $\boldsymbol{\Sigma}' \sim \mathcal{U}^{2\times 2}[-2, 2] \in \mathbb{R}^{2\times 2}$. Here $\mathcal{U}$ means uniform distribution. We then sample 10000 5-way 10-shot tasks for both *meta-training* and *meta-testing*. After every 500 episodes, we sample 500 tasks for *meta-validation*.

**Algorithms.** In this part, we use a ProtoNet [21] trained under $\mathcal{S}/\mathcal{Q}$ protocol as our baseline. It meta-learns a shared embedding function $\phi : \mathbb{R}^2 \to \mathbb{R}^{100}$ across tasks, and classifies an instance into the category of its nearest *support* class center. To be specific, let $\mathbf{c}_n = \frac{1}{K} \sum_{(\mathbf{x}_i, \mathbf{y}_i) \in \mathcal{S} \wedge \mathbf{y}_i = n} \phi(\mathbf{x}_i)$ be the *support* class center of the $n$-th class[4], then for instance $\mathbf{x}$, the model will predict its $N$-dimensional label $\hat{\mathbf{y}}$ as $\hat{y}_n = \frac{\exp\{\langle \mathbf{c}_n, \phi(\mathbf{x}) \rangle\}}{\sum \exp\{\langle \mathbf{c}_n, \phi(\mathbf{x}) \rangle\}}, n \in [N]$. As a comparison, we also train a ProtoNet under $\mathcal{S}/\mathcal{T}$ protocol. Here the target model is constructed by fine-tuning the pre-trained global embedding network on specific tasks. To check whether $\mathcal{S}/\mathcal{T}$ protocol can work with only a few target models, we set the ratio the tasks that have target models to $5\%$ and $10\%$. As presented in last part, we sort all *meta-training* tasks according to their hardness and fine-tune the pre-trained backbone on those hardest tasks. Refer to supplementary material for more details.

**Results and Discussions.** Firstly, we report the *meta-testing* accuracy of different models in Table 4. Methods under $\mathcal{S}/\mathcal{T}$ protocol outperform vanilla ProtoNet by a large margin. Even with only $5\%$ target models, we can obtain a remarkable accuracy improvement. Then, we study why $\mathcal{S}/\mathcal{T}$ protocol can help ProtoNet learn better. In Figure 6, we visualize a 5-way 10-shot task and the decision regions of 3 models in raw 2-d space. Figure 6a is the Bayesian optimal classifier $\mathcal{T}$, *i.e.*, for an instance $\mathbf{x}$, $p(\hat{\mathbf{y}} = n | \mathbf{x}) \propto \frac{1}{2\pi} \frac{1}{|\boldsymbol{\Sigma}_n|^{1/2}} \exp\left\{-\frac{1}{2}(\mathbf{x} - \boldsymbol{\mu}_n)^\top \boldsymbol{\Sigma}_n^{-1}(\mathbf{x} - \boldsymbol{\mu}_n)\right\}$ where $\boldsymbol{\mu}_n$ and $\boldsymbol{\Sigma}_n$ are the mean vector and covariance matrix of class $n$. Because different classes have different covariance matrices, the decision boundary of Bayesian classifier is very steep.

Table 4: Average accuracy on *meta-testing* set. Models trained under $\mathcal{S}/\mathcal{T}$ protocol outperform those trained under $\mathcal{S}/\mathcal{Q}$ protocol even though there are only a few target models. $\phi^{\text{pt}}$ means directly using the pre-trained network to solve *meta-testing* tasks without *meta-training* phase. The second row and the third row represent biased sampling, where we only sample instances that have low likelihoods. When instances are biased, the superiority of $\mathcal{S}/\mathcal{T}$ protocol is more evident because target models make task-specific solvers more robust.

| Protocol | $\phi^{\text{pt}}$ | $\mathcal{S}/\mathcal{Q}$ | $\mathcal{S}/\mathcal{T}$-5% | $\mathcal{S}/\mathcal{T}$-10% |
|---|---|---|---|---|
| ACC | 82.33 | 87.90 | **90.32** | **92.87** |
| ACC($< 0.3$) | 77.41 | 81.25 | **87.66** | **90.14** |
| ACC($< 0.1$) | 65.57 | 70.10 | **79.22** | **84.02** |

Figure 6b and Figure 6c are results of ProtoNet trained under $\mathcal{S}/\mathcal{Q}$ protocol and $\mathcal{S}/\mathcal{T}$ protocol respectively. In Figure 6c, the decision boundary is smooth and regular, which is different from the previous two models. This result offers a natural interpretation of $\mathcal{S}/\mathcal{T}$ protocol's benefit: target models impose a regularization on task-specific solvers, making them more robust to noisy and biased instances. In fact, [32] also gives a similar conclusion: knowledge distillation can be seen as a special label smoothing and it can regularize model training. In order to more clearly verify this property, we sample biased tasks only containing instances that have low likelihoods ($< 0.3$ or $< 0.1$), and test different models on them. In the second row and third row of Table 4, we can see that $\mathcal{S}/\mathcal{T}$ protocol can defend biased sampling to the maximum extent because of the strong supervision offered by target models.

---

[4]With a bit abuse of notation, we use $\mathbf{y}_i = n$ to select instances belonging to the $n$-th class.

Table 5: Average test accuracy with 95% confidence intervals on *meta-testing* tasks of *mini*ImageNet and *tiered*ImageNet. All the methods use ResNet-12 as backbone network except MAML with ⋆ mark. The row with ⋆ mark uses a 4-layer ConvNet as backbone network, which is shallower than ResNet-12. Blue values are cited from existing papers while red values are reproduced by us. Best results are in **bold**. We can see that MAML and ProtoNet trained under $\mathcal{S}/\mathcal{T}$ protocol outperform models trained under $\mathcal{S}/\mathcal{Q}$ protocol even with a few target models. Specifically, ProtoNet trained under $\mathcal{S}/\mathcal{T}$ protocol achieves state-of-the-art performance in most cases.

| Method | *mini*ImageNet | | *tiered*ImageNet | |
|---|---|---|---|---|
| | 5-way 1-shot | 5-way 5-shot | 5-way 1-shot | 5-way 5-shot |
| DeepEMD [33] | $65.91 \pm 0.82$ | $82.41 \pm 0.56$ | $71.16 \pm 0.87$ | $86.03 \pm 0.58$ |
| FEAT [30] | $66.78 \pm 0.20$ | $82.05 \pm 0.14$ | $70.80 \pm 0.23$ | $84.79 \pm 0.16$ |
| FRN [29] | $66.45 \pm 0.19$ | $\mathbf{82.83 \pm 0.13}$ | $72.06 \pm 0.22$ | $86.89 \pm 0.14$ |
| MAML ($\mathcal{S}/\mathcal{Q}$)⋆ [3] | $48.70 \pm 1.84$ | $63.11 \pm 0.92$ | - | - |
| MAML ($\mathcal{S}/\mathcal{Q}$, re-implement) | $58.84 \pm 0.25$ | $74.62 \pm 0.38$ | $63.02 \pm 0.30$ | $67.26 \pm 0.32$ |
| MAML ($\mathcal{S}/\mathcal{T}$-5%) | $59.14 \pm 0.33$ | $75.77 \pm 0.29$ | $64.52 \pm 0.30$ | $68.39 \pm 0.34$ |
| MAML ($\mathcal{S}/\mathcal{T}$-10%) | $60.06 \pm 0.35$ | $76.34 \pm 0.42$ | $65.23 \pm 0.45$ | $70.02 \pm 0.33$ |
| ProtoNet ($\mathcal{S}/\mathcal{Q}$) [21] | $60.37 \pm 0.83$ | $78.02 \pm 0.57$ | $65.65 \pm 0.92$ | $83.40 \pm 0.65$ |
| ProtoNet ($\mathcal{S}/\mathcal{Q}$, re-implement) | $65.30 \pm 0.30$ | $79.93 \pm 0.39$ | $70.34 \pm 0.45$ | $84.68 \pm 0.55$ |
| ProtoNet ($\mathcal{S}/\mathcal{T}$-5%) | $67.35 \pm 0.49$ | $81.67 \pm 0.62$ | $71.25 \pm 0.37$ | $85.80 \pm 0.31$ |
| ProtoNet ($\mathcal{S}/\mathcal{T}$-10%) | $\mathbf{68.03 \pm 0.52}$ | $82.53 \pm 0.47$ | $\mathbf{72.41 \pm 0.39}$ | $\mathbf{86.91 \pm 0.47}$ |

Table 6: Ablation study. "Random" means selecting tasks randomly rather than according to their hardness. "$\phi^{pt}$" means using the pre-trained network as target model for all tasks. Best results are in **bold**. We can see that our proposed heuristic hardness metric and the fine-tuning strategy improve model performance.

| Model | *mini*ImageNet | | *tiered*ImageNet | |
|---|---|---|---|---|
| | 5-way 1-shot | 5-way 5-shot | 5-way 1-shot | 5-way 5-shot |
| MAML ($\mathcal{S}/\mathcal{Q}$) | 58.84 | 74.62 | 63.02 | 67.26 |
| MAML ($\mathcal{S}/\mathcal{T}$-10%-random) | 59.66 | 74.90 | 65.11 | 68.63 |
| MAML ($\mathcal{S}/\mathcal{T}$-10%-$\phi^{pt}$) | 59.35 | 75.88 | 64.78 | 69.26 |
| MAML ($\mathcal{S}/\mathcal{T}$-10%-hardness-$\phi^{ft}$) | **60.06** | **76.34** | **65.23** | **70.02** |
| ProtoNet ($\mathcal{S}/\mathcal{Q}$) | 65.30 | 79.93 | 70.34 | 84.68 |
| ProtoNet ($\mathcal{S}/\mathcal{T}$-10%-random) | 66.72 | 81.05 | 71.22 | 85.37 |
| ProtoNet ($\mathcal{S}/\mathcal{T}$-10%-$\phi^{pt}$) | 67.47 | 81.70 | 71.55 | 86.04 |
| ProtoNet ($\mathcal{S}/\mathcal{T}$-10%-hardness-$\phi^{ft}$) | **68.03** | **82.53** | **72.41** | **86.91** |

## 5.4 Empirical Study: Benchmark Evaluation

In this part, we evaluate our $\mathcal{S}/\mathcal{T}$ protocol on two benchmark datasets, *i.e.*, *miniImageNet* [24] and *tiered*ImageNet [17]. Refer to supplementary material for dataset details.[5] We try to answer four questions: (1) Can we achieve SOTA performance with a classic meta-learning model trained under $\mathcal{S}/\mathcal{T}$ protocol? (2) How does each component influence model's performance? (3) How does the hyper-parameter $\lambda$ influence model's performance? (4) How much time does $\mathcal{S}/\mathcal{T}$ protocol cost?

**Algorithms.** We implement two classic meta-learning algorithms, MAML and ProtoNet, under $\mathcal{S}/\mathcal{T}$ protocol. We use ResNet-12 as the backbone network, which is pre-trained on the *meta-training* set. For a fair comparison, we only include other algorithms that also use ResNet-12 as backbone network in Table 5. More implementation details can be found in the supplementary material.

**Competitive Results against SOTA.** We show in Table 5 that MAML or ProtoNet can be improved a lot when trained under $\mathcal{S}/\mathcal{T}$ protocol with only 5% or 10% target models. Note that vanilla ProtoNet does not use pre-training trick, and we re-implement it with pre-training. ProtoNet is proposed in 2017, but we can obtain SOTA performance by retraining it under $\mathcal{S}/\mathcal{T}$ protocol with only a few target models. This verifies the superiority of $\mathcal{S}/\mathcal{T}$ protocol. In fact, $\mathcal{S}/\mathcal{T}$ protocol is a generic training protocol that can be applied to any meta-learning algorithm, and we apply $\mathcal{S}/\mathcal{T}$ protocol to more meta-learning algorithms in the supplementary material to show the effectiveness of our method.

---

[5]Our code is available at `https://github.com/njulus/ST`.

Table 7: Time consumption for fine-tuning target models on *mini*ImageNet.

| Number of Target Models | Time Consumption (min) |
|:---:|:---:|
| 500 (5%) | 224.3 |
| 1000 (10%) | 420.6 |
| 2000 (20%) | 851.7 |

**Ablation Study.** In this part, we check the effectiveness of each component. Table 6 shows that our proposed hardness metric and fine-tuning strategy help to improve performance. Randomly sampling $10\%$ tasks and constructing target models for these tasks improves model performance. The third row and the seventh row in Table 6 verify that learning from target models is beneficial even though the target models are not optimal. With only $10\%$ locally fine-tuned target models and our heuristic hardness metric, we can achieve nearly state-of-the-art performance by ProtoNet.

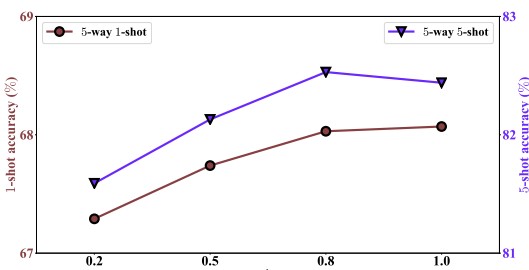

Figure 7: Performance change over $\lambda$. We can see that larger $\lambda$ tends to benefit model accuracy. We set $\lambda$ to $0.8$, a relatively large value.

**Hyper-Parameter.** We check the influence of hyper-parameter $\lambda$ in Equ (7). We sample 5-way tasks from *mini*ImageNet, and try different $\lambda$ values. Figure 7 shows that larger $\lambda$ tends to benefit model performance. We set $\lambda$ to $0.8$, a relatively large value, in most of experiments.

**Time Consumption.** In $\mathcal{S}/\mathcal{T}$ protocol for few-shot learning, we need to construct target models through fine-tuning the globally pre-trained network. This will cost extra time to train a model. In this part, we try to answer the following question: how much time does $\mathcal{S}/\mathcal{T}$ protocol cost in few-shot learning? We range the ratio of tasks that have target models in $\{5\%, 10\%, 20\%\}$, and report the time consumption of fine-tuning target models on *mini*ImageNet. Results are shown in Table 7. We run the experiment on an Nvidia GeForce RTX 2080ti GPU and Intel(R) Xeon(R) Silver 4110 CPU. We can see that about $4$ hours are needed to fine-tune target models for $5\%$ *meta-training* tasks, and time consumption for fine-tuning $2000$ target models is still acceptable.

## 6   Conclusion

In this paper, we study $\mathcal{S}/\mathcal{T}$ meta-learning protocol that evaluates a task-specific solver by comparing it to a target model. $\mathcal{S}/\mathcal{T}$ protocol offers a more informative supervision signal for meta-learning, but is difficult to use in practice owing to its high computational cost. We find that by only deploying target models for those hardest tasks, we can improve existing meta-learning algorithms while maintaining efficiency. We propose a heuristic task hardness metric and a convenient target model construction method for few-shot learning. Experiments on synthetic datasets and benchmark datasets demonstrate the superiority of $\mathcal{S}/\mathcal{T}$ protocol and effectiveness of our proposed method.

## Acknowledgements

This work is supported by National Key R&D Program of China (2020AAA0109401), NSFC (41901270, 61773198, 62006112), NSF of Jiangsu Province (BK20190296, BK20200313), and CCF-Baidu Open Fund (NO.2021PP15002000).

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
