# Supplementary Material of Towards Enabling Meta-Learning from Target Models

**Su Lu**   **Han-Jia Ye**   **Le Gan**   **De-Chuan Zhan**
State Key Laboratory for Novel Software Technology
Nanjing University, Nanjing, 210023, China
{lus,yehj}@lamda.nju.edu.cn, {ganl,zhandc}@nju.edu.cn

## Abstract

This is the supplementary material of paper "Towards Enabling Meta-Learning from Target Models". We give implementation details, more discussions, and more experiment results in this material.

## 1   Sinusoid Regression

In Section 4.2 of the main body, we construct a synthetic regression problem to verify the effectiveness of $\mathcal{S}/\mathcal{T}$ protocol. In this experiment, we assume that target models for all *meta-training* tasks are available, and show that learning from target models can offer more supervision information to the meta-model. This section gives more details about this experiment.

### 1.1   Dataset Generation

A sinusoid regression task is defined as $\mathcal{T}(x) = a\sin(bx - c)$. Here we use symbol $\mathcal{T}$ to represent both the sinusoid function itself and the target model corresponding to each task. In other words, we assume that "true" target models are accessible in this experiment.

We randomly sample 10000 tasks for *meta-training* and 10000 tasks for *meta-testing*. We sample 200 tasks for *meta-validation* for every 200 *meta-training* tasks. To get tasks that come from a same distribution, we uniformly sample $a$, $b$, and $c$ from $[0.1, 5]$, $[0.5, 2]$, and $[0.5, 2\pi]$ respectively. In each task, we sample 10 *support* instances for both $\mathcal{S}/\mathcal{Q}$ and $\mathcal{S}/\mathcal{T}$ protocol, and sample 30 *query* instances for $\mathcal{S}/\mathcal{Q}$ protocol. The instance sampling procedure is as follows: uniformly sampling $x$ in range $[-5, 5]$ and set $y = \mathcal{T}(x) + \epsilon$ where $\epsilon \sim \mathcal{N}(0, 0.5)$.

### 1.2   Models and Algorithms

MAML [1] and ProtoNet [5] are two classic meta-learning algorithms. While MAML can be directly applied in regression problems, ProtoNet is originally designed for classification problems. In this experiment, we modify ProtoNet slightly to fit regression problem. In detail, we try to meta-learn an embedding function $\phi : \mathbb{R} \to \mathbb{R}^{100}$, with assistance of which the similarity-based regression model $g(\cdot\,; \{\phi(x_i)|(x_i, y_i) \in \mathcal{S}\})$ works well across all tasks. The embedding network $\phi$ is implemented as an MLP, and we illustrate its structure in Figure 1. In this model, for any instance $(x, y)$, predicted label is given by $\hat{y} = g(x) = \sum_{(x_i, y_i) \in \mathcal{S}} w_i y_i$ and $w_i = \frac{\exp\{\langle \phi(x_i), \phi(x) \rangle\}}{\sum \exp\{\langle \phi(x_i), \phi(x) \rangle\}}$. A same embedding network is used in two algorithms. We train MAML and ProtoNet under $\mathcal{S}/\mathcal{Q}$ protocol and $\mathcal{S}/\mathcal{T}$ protocol. When using $\mathcal{S}/\mathcal{Q}$ protocol, we minimize MSE loss on 30 *query* instances to optimize $\phi$. For $\mathcal{S}/\mathcal{T}$ protocol, we match the solver to the target model in output

35th Conference on Neural Information Processing Systems (NeurIPS 2021).

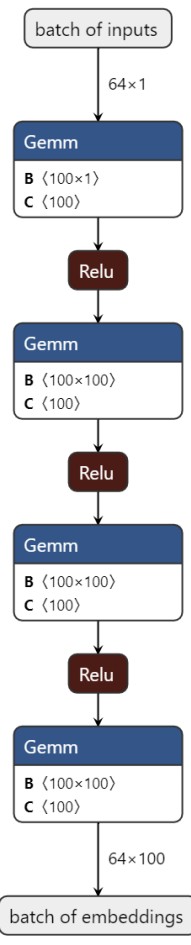

Figure 1: Structure of embedding network $\phi$ used in sinusoid regression. Batch size is set to $64$.

space, and set $\mathbf{D}(\mathcal{T}(x_i), g(x_i)) = \|\mathcal{T}(x_i) - g(x_i)\|_2^2$. Thus, the loss function under $\mathcal{S}/\mathcal{T}$ protocol is $\mathcal{L}(g, \mathcal{T}) = \sum_{(x_i, y_i) \in \mathcal{S}^{tr}} \left[ (1 - \lambda)\|g(x_i) - y_i\|_2^2 + \lambda\|g(x_i) - \mathcal{T}(x_i)\|_2^2 \right]$. $\lambda$ is a hyper-parameter.

## 1.3 Implementation Details

Hyper-parameter $\lambda$ is set to $0.5$ by default, and Table 2 in the main body is an ablation study about $\lambda$. For both MAML and ProtoNet, we use SGD optimizer to train our network. The initial learning rate is set to $0.01$, which decreases by $0.8$ after training on $4000$, $6000$, and $8000$ tasks. The weight decay and momentum of SGD optimizer is set to $0.0005$ and $0.9$ respectively.

## 1.4 Denoising Effect

We can show that the meta-learning loss under $\mathcal{S}/\mathcal{T}$ protocol $(1 - \lambda)\|g(x) - y\|_2^2 + \lambda\|g(x) - \mathcal{T}(x)\|_2^2$ is an upper bound of $\|g(x) - (y - \lambda\epsilon)\|_2^2$, which is the standard MSE loss between the output of

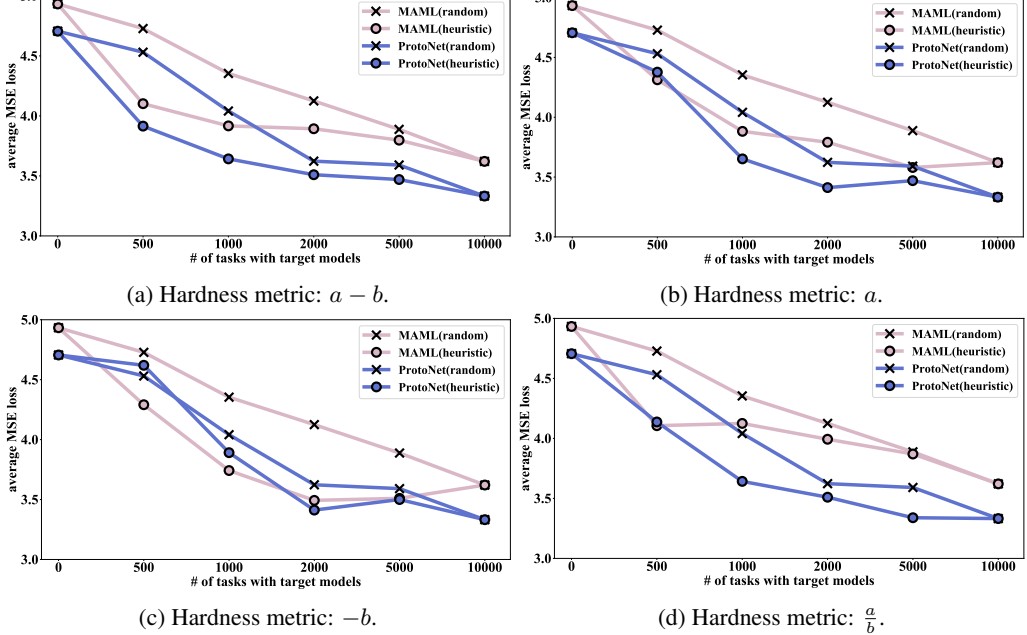

(a) Hardness metric: $a - b$.

(b) Hardness metric: $a$.

(c) Hardness metric: $-b$.

(d) Hardness metric: $\frac{a}{b}$.

Figure 2: Change of MSE loss over number of *meta-training* tasks that have target models. By selecting hard tasks heuristically, we are able to obtain an evident performance gain with a small number of target models.

solver $g$ and cleaner label $y - \lambda\epsilon$ (raw label $y$ equals to $\mathcal{T}(x) + \epsilon$). In detail,

$$
\begin{aligned}
& (1 - \lambda)\|g(x) - y\|_2^2 + \lambda\|g(x) - \mathcal{T}(x)\|_2^2 \\
= {} & (1 - \lambda)\|g(x) - y\|_2^2 + \lambda\|g(x) - (y - \epsilon)\|_2^2 \\
= {} & (1 - \lambda)\left([g(x)]^2 + y^2 - 2yg(x)\right) + \lambda\left([g(x)]^2 + (y - \epsilon)^2 - 2(y - \epsilon)g(x)\right) \\
= {} & [g(x)]^2 + (1 - \lambda)y^2 - 2(1 - \lambda)yg(x) + \lambda(y^2 + \epsilon^2 - 2\epsilon y) - 2\lambda(y - \epsilon)g(x) \\
= {} & [g(x)]^2 + y^2 - 2yg(x) + 2\lambda yg(x) + \lambda\epsilon^2 - 2\lambda\epsilon y - 2\lambda yg(x) + 2\lambda\epsilon g(x) \\
= {} & [g(x)]^2 + y^2 - 2yg(x) + \lambda\epsilon^2 - 2\lambda\epsilon y + 2\lambda\epsilon g(x) \\
= {} & [g(x)]^2 - 2g(x)(y - \lambda\epsilon) + (y^2 + \lambda\epsilon^2 - 2\lambda\epsilon y) \\
\geq {} & [g(x)]^2 - 2g(x)(y - \lambda\epsilon) + (y^2 + \lambda^2\epsilon^2 - 2\lambda\epsilon y) \\
= {} & [g(x)]^2 - 2g(x)(y - \lambda\epsilon) + (y - \lambda\epsilon)^2 \\
= {} & \|g(x) - (y - \lambda\epsilon)\|_2^2
\end{aligned}
\tag{1}
$$

The equality holds when $\lambda$ equals to 0 or 1. In these two cases, the $\mathcal{S}/\mathcal{T}$ loss degenerates to $\mathcal{S}/\mathcal{Q}$ loss or target model loss.

### 1.5 Hardness Metric

In Figure 2a (same as Figure 4 in the main body), we visualize the change of MSE loss over number of *meta-training* tasks that have target models. We can see that only a small number of target models can benefit model performance. In this part, we further try other hardness metric. We use $a$, $-b$, $\frac{a}{b}$ as hardness metrics, and visualize the results in Figure 2b, Figure 2c, and Figure 2d respectively. We can see that all of these heuristic metrics successfully help the selection of hard tasks to some extent.

### 1.6 Visualization

We give visualization of more *meta-testing* tasks in Figure 3. Models trained under $\mathcal{S}/\mathcal{T}$ protocol can fit the target curves better than models trained under $\mathcal{S}/\mathcal{Q}$ protocol.

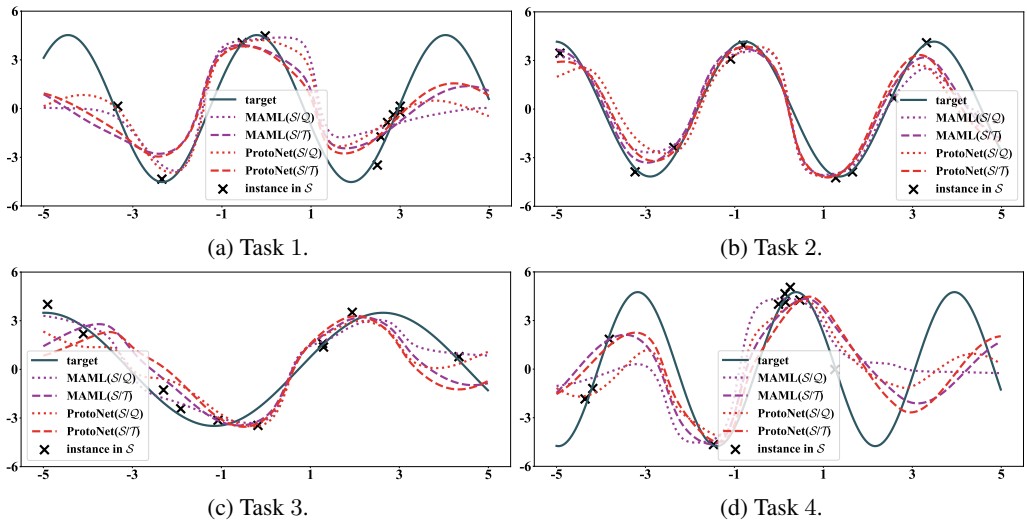

(a) Task 1.   (b) Task 2.

(c) Task 3.   (d) Task 4.

Figure 3: Visualization of $4$ randomly sampled *meta-testing* tasks. Dotted lines are used for $\mathcal{S}/\mathcal{Q}$ protocol while dashed lines are used for $\mathcal{S}/\mathcal{T}$ protocol.

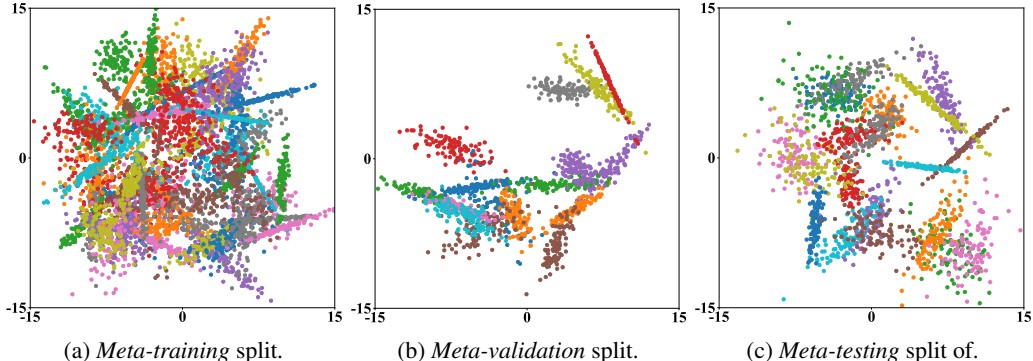

(a) *Meta-training* split.   (b) *Meta-validation* split.   (c) *Meta-testing* split of.

Figure 4: Visualization of Gaussian dataset.

## 2  Gaussian Classification

This section gives more details about the Gaussian classification problem discussed in Section 5.3 of the main body.

### 2.1  Dataset Generation

In this experiment, we randomly generate $100$ 2-d Gaussian distributions. There are $64$ classes for *meta-training*, 16 classes for *meta-validation*, and 20 classes for *meta-testing*. We sample $100$ instances for each class to form the whole dataset. For each class, we sample its mean vector $\boldsymbol{\mu} \sim \mathcal{U}^2[-10, 10] \in \mathbb{R}^2$ and covariance matrix $\boldsymbol{\Sigma} = \boldsymbol{\Sigma}'^{\top}\boldsymbol{\Sigma}'$ where $\boldsymbol{\Sigma}' \sim \mathcal{U}^{2\times2}[-2, 2] \in \mathbb{R}^{2\times2}$. Here $\mathcal{U}$ means uniform distribution. *Meta-training* set, *meta-validation* set, and *meta-testing* set are shown in Figure 4a, Figure 4b, and Figure 4c respectively. We then sample $10000$ 5-way 10-shot tasks for both *meta-training* and *meta-testing*. After every 500 episodes, we sample 500 tasks for *meta-validation*.

### 2.2  Models and Algorithms

In this part, we use a ProtoNet [5] trained under $\mathcal{S}/\mathcal{Q}$ protocol as our baseline. It meta-learns a shared embedding function $\phi : \mathbb{R}^2 \rightarrow \mathbb{R}^{100}$ across tasks, and classifies an instance into the category of its nearest *support* class center. The structure of $\phi$ is visualized in Figure 5.

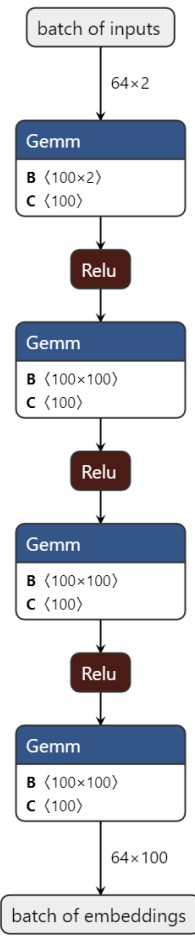

Figure 5: Structure of embedding network $\phi$ used in Gaussian classification. Batch size is set to $64$.

Let $\mathbf{c}_n = \frac{1}{K} \sum_{(\mathbf{x}_i, \mathbf{y}_i) \in \mathcal{S} \wedge \mathbf{y}_i = n} \phi(\mathbf{x}_i)$ be the *support* class center of the $n$-th class[1], then for instance $\mathbf{x}$, the model will predict its $N$-dimensional label $\hat{\mathbf{y}}$ as $\hat{y}_n = \frac{\exp\{\langle \mathbf{c}_n, \phi(\mathbf{x}) \rangle\}}{\sum \exp\{\langle \mathbf{c}_n, \phi(\mathbf{x}) \rangle\}}, n \in [N]$. As a comparison, we also train a ProtoNet under $\mathcal{S}/\mathcal{T}$ protocol. Here the target model is constructed by fine-tuning the pre-trained global embedding network on specific tasks. To check whether $\mathcal{S}/\mathcal{T}$ protocol can work with only a few target models, we set the ratio the tasks that have target models to $5\%$ and $10\%$. As presented in last part, we sort all *meta-training* tasks according to their hardness and fine-tune the pre-trained backbone on those hardest tasks.

### 2.3 Implementation Details

Hyper-parameter $\lambda$ is set to $0.8$ by default. For both $\mathcal{S}/\mathcal{Q}$ protocol and $\mathcal{S}/\mathcal{T}$ protocol, we use SGD optimizer to train ProtoNet. The backbone network is pre-trained on the whole *meta-training* set using cross-entropy loss. The initial learning rate is set to $0.001$, which decreases by $0.8$ after training on $4000$, $6000$, and $8000$ tasks. The weight decay and momentum of SGD optimizer is set to $0.0005$ and $0.9$ respectively.

### 2.4 Visualization

We give visualization of more *meta-testing* tasks in Figure 6. Models trained under $\mathcal{S}/\mathcal{T}$ protocol have smooth classification boundaries and are more robust to biased and noisy instances.

---

[1]With a bit abuse of notation, we use $\mathbf{y}_i = n$ to select instances belonging to the $n$-th class.

Table 1: Average accuracy of different models on biased tasks. Models trained under $\mathcal{S}/\mathcal{T}$ protocol outperform models trained under $\mathcal{S}/\mathcal{Q}$ protocol due to the regularization effect.

| Protocol | $\phi^{\mathrm{pt}}$ | $\mathcal{S}/\mathcal{Q}$ | $\mathcal{S}/\mathcal{T}$-5% | $\mathcal{S}/\mathcal{T}$-10% |
|---|---|---|---|---|
| Accuracy | 82.33 | 87.90 | **90.32** | **92.87** |
| Accuracy (<0.7) | 81.25 | 86.47 | **88.90** | **91.33** |
| Accuracy (<0.5) | 79.69 | 84.50 | **87.72** | **90.58** |
| Accuracy (<0.3) | 77.41 | 81.25 | **87.66** | **90.14** |
| Accuracy (<0.1) | 65.57 | 70.10 | **79.22** | **84.02** |

## 2.5 Biased Sampling

In Table 4 of the main body, we study the influence of $\mathcal{S}/\mathcal{T}$ protocol when sampled data points are biased. Specifically, we sample biased tasks only containing data points that have low likelihoods ($<$ 0.3 or $< 0.1$), and show that models trained under $\mathcal{S}/\mathcal{T}$ protocol outperform models trained under $\mathcal{S}/\mathcal{Q}$ protocol due to the regularization effect of $\mathcal{S}/\mathcal{T}$ protocol. In this part, we give the experiment results of different likelihood thresholds in Table 1, and verify our claims again.

## 3 Benchmark Evaluation

We also study an application case, few-shot learning, on two widely used benchmark datasets. In this part, we give detailed description of datasets, implementation details, and more experiment results.

### 3.1 Dataset Description

*Mini*ImageNet [6] and *tiered*ImageNet [3] are two widely used benchmark datasets in few-shot learning. *Mini*ImageNet dataset was firstly proposed by [6] and it is a subset of ILSVRC-12 [4]. In this dataset, there are 100 classes and 600 images in each class. Each image in *mini*ImageNet is resized to $84 \times 84$. We follow [2] to split *mini*ImageNet, which means the total 100 classes are divided into *meta-training* set, *meta-validating* set, and *meta-testing* set, with 64, 16, and 20 classes respectively. *Tiered*ImageNet is a larger subset of ILSVRC-12. There are 608 classes and 779165 images in total. These classes are divided into 34 categories, with each category containing between 10 to 30 classes. Images in *tiered*ImageNet are also resized to $84 \times 84$. Following [3], we split *tiered*ImageNet into *meta-training*, *meta-validating* and *meta-testing* set, with 20, 6, and 8 categories respectively.

### 3.2 Implementation Details

In benchmark evaluation, we use ResNet-12 as backbone network for MAML, ProtoNet, and other comparison algorithms. The structure of ResNet-12 is shown is Figure 7. The backbone network is pre-trained on *meta-training* split using cross-entropy loss. We utilize data augmentation in pre-training phase. In detail, each image is randomly resized and cropped to $84 \times 84$, and then horizontally flipped with a probability 0.5. Finally, images are normalized with mean $[0.485, 0.456, 0.406]$ and standard deviation $[0.229, 0.224, 0.225]$. In *meta-training* and *meta-testing* phase, we only center crop and normalize the images. Number of *meta-training* episodes and *meta-testing* episodes are both 10000. We optimize our model using SGD optimizer on 10000 tasks. The momentum and weight decay of the optimizer are set to 0.9 and 0.0005 respectively. The initial learning rate for the pre-trained embedding network and other modules are set to 0.001 and 0.01 respectively. Two learning rates are decreased by 0.8 after every 2000 episodes. Hyper-parameter $\lambda$ is set to 0.8 by default. When constructing target model for a specific task, we fine-tune the pre-trained network on all *meta-training* instances that belong to the corresponding classes for 3 epochs. The learning rate in fine-tuning phase is set to 0.0002.

### 3.3 Other Models Trained under $\mathcal{S}/\mathcal{T}$ Protocol

In the main body of this paper, we mainly apply $\mathcal{S}/\mathcal{T}$ protocol to two well-known meta-learning algorithms, *i.e.*, MAML and ProtoNet, and show that with only a small number of target models,

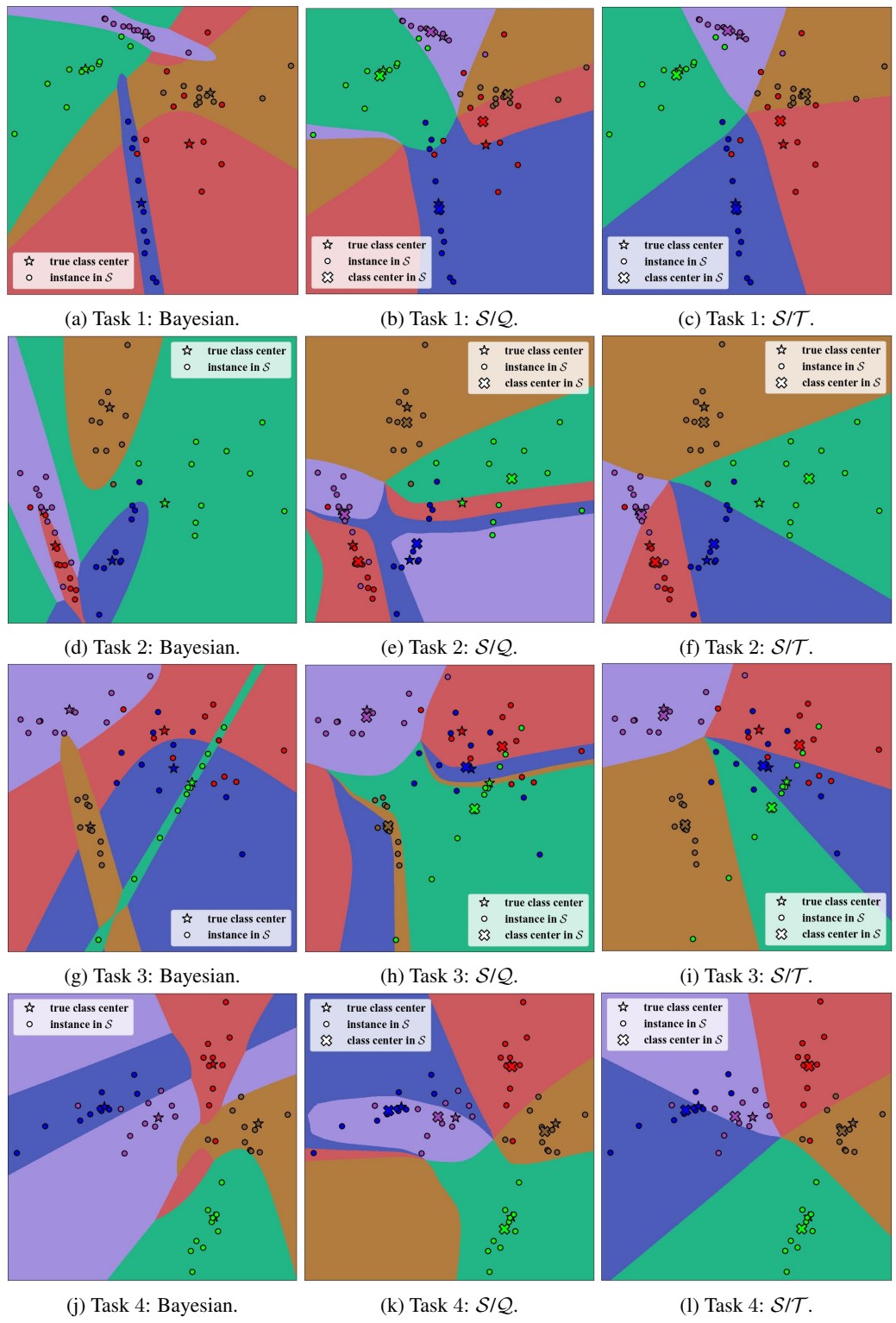

Figure 6: Decision boundaries of three different models in raw 2-d space. Four tasks are randomly sampled. Different point colors represent different classes, and different background colors represent different classification regions.

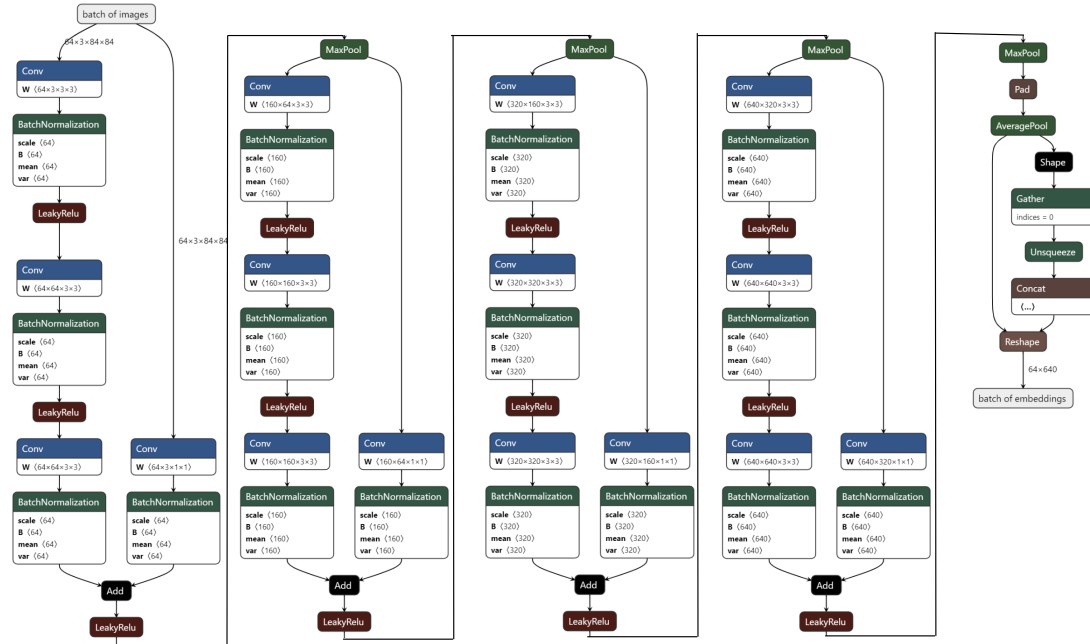

Figure 7: Structure of embedding network $\phi$ used in benchmark evaluation. Batch size is set to $64$.

Table 2: Average test accuracy with $95\%$ confidence intervals on *meta-testing* tasks of *mini*ImageNet. We use ResNet-12 as backbone network. $\mathcal{S}/\mathcal{T}$ protocol improves the performance of FEAT even though we only have a small number of target models.

| Method | *mini*ImageNet | |
| --- | --- | --- |
| | 5-way 1-shot | 5-way 5-shot |
| FEAT($\mathcal{S}/\mathcal{Q}$) [7] | $66.78 \pm 0.20$ | $82.05 \pm 0.14$ |
| FEAT($\mathcal{S}/\mathcal{T}$-5%) | $67.32 \pm 0.41$ | $81.60 \pm 0.38$ |
| FEAT($\mathcal{S}/\mathcal{T}$-10%) | $\mathbf{68.23 \pm 0.37}$ | $\mathbf{82.53 \pm 0.42}$ |

$\mathcal{S}/\mathcal{T}$ protocol can improve classic meta-learning algorithms. In this part, we try to train a FEAT [7] under $\mathcal{S}/\mathcal{T}$ protocol, and check whether $\mathcal{S}/\mathcal{T}$ protocol can improve SOTA algorithms like FEAT. Table 2 shows the results. We can see that $\mathcal{S}/\mathcal{T}$ protocol also improves the performance of FEAT. However, the performance gap between ProtoNet and FEAT is decreased when we train them under $\mathcal{S}/\mathcal{T}$ protocol. FEAT trained under $\mathcal{S}/\mathcal{Q}$ protocol outperforms ProtoNet trained under $\mathcal{S}/\mathcal{Q}$ protocol by $1.48\%$, but FEAT trained under $\mathcal{S}/\mathcal{T}$ protocol gets a similar accuracy to that of ProtoNet trained under $\mathcal{S}/\mathcal{T}$ protocol. This is because target models offer more supervision information, so that the effect of improvement on model and algorithm is weakened.