# OpenReview forum: "Towards Enabling Meta-Learning from Target Models"
_NeurIPS.cc/2021/Conference — NeurIPS 2021 Poster_

### Official Review · Reviewer_mVfo · 2021-07-04

**Rating:** 6
**Confidence:** 2

**Summary:**

This paper proposes an efficient S/T protocol meta-learning algorithm. Specifically, in order to reduce the number of required target models and the high computational cost, only the target models on those hardest tasks are constructed by fine-tunning the pre-trained network, then the knowledge distillation is used to match the task-specific solvers to target models in the output space. Experiment results verify the effectiveness of the proposed algorithms.


**Limitations And Societal Impact:**

No potential negative societal impact of their work

**Main Review:**

This paper proposes an efficient S/T protocol meta-learning algorithm. Specifically, in order to reduce the number of required target models and the high computational cost, only the target models on those hardest tasks are constructed by fine-tunning the pre-trained network, then the knowledge distillation is used to match the task-specific solvers to target models in the output space. Experiment results verify the effectiveness of the proposed algorithms.

Strong points:
1.The motivation is clear.

2.An efficient S/T protocol meta learning algorithms is proposed by fine-tuning globally pre-trained model on several hardest tasks.

3.Experiment results illustrate that the classic meta learning algorithms trained via the proposed S/T protocol can achieve SOTA performance.

Weak points:
1.The main contribution should be emphasized.  In this manuscript, the two contributions are introduced and verified respectively. It would be better to emphasize on the intro section.

2.The proposed algorithm is still under the condition of same task distribution, because the target models is always trained on a sufficiently informative dataset. However, in many real-world scenarios the training distribution differs from the test distribution. How the out of distribution will influence the algorithm performance should be further investigated in several benchmark datasets, such as [1].

3.Meta learning algorithms aim to achieve fast adaptive ability during training phase. Thus, when encountered new tasks, even those tasks have new label space, the learnt model can still achieve better performance with only few data. If the label space of new tasks is different from the training task one, will the proposed approach still effective?

4.The recognition of task hardness depends on the base model \phi^{pt}, which is pre-trained on training task dataset. Will the per-trained model influence the final performance should be further investigated.
5.It would be better to give in-depth analysis on why this algorithm can generalize better.


Overall, the topic is interesting, and experiments verified the proposed approach outperform than other algorithms. However, the contribution is still marginal, and this research lacks theoretical analysis. Authors didn’t provide insight into how the proposed method can achieve better generalizability.

If providing satisfying answers, I will accept the paper.



[1] Koh, P. W., Sagawa, S., Marklund, H., Xie, S. M., Zhang, M., Balsubramani, A., ... & Liang, P. (2020). Wilds: A benchmark of in-the-wild distribution shifts. arXiv preprint arXiv:2012.07421.

**Time Spent Reviewing:**

at least 12 hours

---

> ### Author Response · Authors · 2021-08-10
> **Official Comment in Response to Reviewer mVfo (Part I)**
>
> [*This is the first part of our responses.*]
>
> Dear reviewer,
>
> thank you for your detailed comments! We appreciate your positive remarks like 'the motivation is clear' and '$\mathcal{S}$/$\mathcal{T}$ protocol is efficient'. Here are our responses.
>
> ## Contribution of our work
> The reviewer mentioned that 'the main contribution should be emphasized', and we agree with you. Now we reiterate our contributions, and we will plug them into introduction in the final version. Thanks for your suggestion! Our contributions are three-fold:
> - Novel Idea: Although $\mathcal{S}$/$\mathcal{Q}$ protocol and $\mathcal{S}$/$\mathcal{T}$ protocol both appeared in year 2016, $\mathcal{S}$/$\mathcal{T}$ protocol is almost overlooked by meta-learning society owing to its high computational cost. Thus, a key contribution of this paper is looking into this special training style and trying to put it into practice. The novelty of our work is also pointed out by Reviewer-f3eh and Reviewer-V6R5.
> >     *Reviewer-f3eh*: Very novel, and surprising work. This could be the beginning of a good meta learning penalty term.
> >     *Reviewer-V6R5*: This paper can to some extent advance meta-learning research.
>
> - Practical Method: We tackle the computational burden of constructing target models in $\mathcal{S}$/$\mathcal{T}$ protocol, and propose an easy and effective method to construct target models.
> - Convincing Results: We empirically verify the effectiveness of target models, hardness metric, and model construction method. Two classic meta-learning algorithms (MAML[1] and ProtoNet[2]) achieve remarkable performance gain when trained under $\mathcal{S}$/$\mathcal{T}$ protocol. Reviewer-V6R5 also considered our work as sufficient and solid.
> >     *Reviewer-V6R5*: The study and analysis on the target models are sufficient and solid.
>
> ## Same task distribution v.s. different task distributions
> The reviewer mentioned that 'the proposed algorithm is still under the condition of same task distribution', and this is true. Actually, we evaluate the meta-model on same-task distribution following the widely adopted meta-learning setting[1,2,3,9,10]
>
> However, the problem raised by reviewer is very interesting! We will investigate how distribution gap can influence meta-learning algorithm. In detail, we try to meta-learn a meta-model on a task distribution $\mathcal{P}^{\text{tr}}$, and test it on tasks sampled from another task distribution $\mathcal{P}^{\text{ts}}$.
>
> Actually, there exist several papers that tackle this problem[6,7] in few-shot learning scenario, i.e., learning on a domain and testing on few-shot tasks sampled from another domain. This setting is called Cross-Domain Few-Shot Classification in current literatures. [6] proposed to use feature-wise transform to encourage learning representations with improved ability to generalize, and [7] proposed a benchmark for cross-domain few-shot classification that consists of images from a diverse assortment of image acquisition methods.
>
> Now we perform two groups of experiments. In the first group, we strictly follow the experiment setting in [6], and check whether $\mathcal{S}$/$\mathcal{T}$ protocol can improve the generalization ability of meta-model. In the second group, we use Wilds dataset[5] to validate the effect of $\mathcal{S}$/$\mathcal{T}$ protocol.
>
> ### Experiment on miniImageNet and CUB
> MiniImageNet[3] and CUB[4] are used in [6]. We meta-learn a meta-model on miniImageNet, and then test it on CUB. In this part, we use MatchingNet[3] and ProtoNet[2] as baseline algorithms, and show the effectiveness of special training protocl. All testing tasks are 5-way 1-shot, and other detailed experiment setting are same as those in Section 5.4. Mean accuracies are shown in the following table.
>
> | Method | 5-way 1-shot acc on CUB |
> | :---- | :----: |
> | MatchNet ($\mathcal{S}$/$\mathcal{Q}$ protocol) | 37.90 (cited from [6]) |
> | MatchNet + FT ($\mathcal{S}$/$\mathcal{Q}$ protocol) | 41.74 (cited from [6])|
> | MatchNet + LFT ($\mathcal{S}$/$\mathcal{Q}$ protocol) | 43.29 (cited from [6]) |
> | MatchNet ($\mathcal{S}$/$\mathcal{T}$ protocol) | 40.33 |
> | ProtoNet ($\mathcal{S}$/$\mathcal{Q}$ protocol) | 39.27 |
> | ProtoNet ($\mathcal{S}$/$\mathcal{T}$ protocol) | 42.40 |
>
> The second row and the third row are special feature transformations designed for cross-domain few-shot learning in [6]. We can see that for both MatchNet and ProtoNet, models trained under $\mathcal{S}$/$\mathcal{T}$ protocol outperform those trained under $\mathcal{S}$/$\mathcal{Q}$ protocol. Moreover, using a MatchNet or ProtoNet without speicially designed transformations (fourth row and sixth row) under $\mathcal{S}$/$\mathcal{T}$ protocol, we can achieve competitive results against MatchNet + FT (second row). This means $\mathcal{S}$/$\mathcal{T}$ protocol enhances meta-model's generalization ability across domains.
>
> ### Experiment on Wilds benchmark
> Wilds[5] is a benchmark of in-the-wild distribution shifts spanning diverse data modalities and applications, from tumor identification to wildlife monitoring to poverty mapping. It contains 10 datasets, and each dataset contains domain gap between training split and testing split. There exist some papers that run domain adaptation algorithms on Wilds, but unfortunately, there is not any paper that conduct few-shot learning or meta-learning experiment on Wilds. Thus, we need to determine the problem setting by ourselves.
>
> Due to time limit, we download one relatively small dataset, RxRx1[8], from Wilds benchmark, and perform experiments on RxRx1. RxRx1 contains images of cellular morphological variation from different experimental batches. Images in training set and images in testing set are from different experiment executions, and batch effects bring a domain gap between training data and testing data.
>
> Now we describe detailed experiment settings. RxRx1 contains 1139 classes, and each class stands for a specific genetic treatment. To mimic cross-domain few-shot learning setting, we need to ensure:
> - Meta-training split and meta-testing split contain different classes.
> - Meta-training split and meta-testing split are from different domains.
>
> Thus, we firstly extract 60% classes from raw training set as meta-training split, and then extract non-overlapping 20% classes from raw validation set as meta-validation split, and finally treat 20% remaining classes from raw testing set as meta-testing split. Following [5], we use a ResNet-50 as backbone network. We use ProtoNet as a representative meta-learning algorithm, and report 5-way 1-shot accuracy and 5-way 5-shot accuracy on meta-testing split. The mean accuracies are shown in the following table. We can see that model trained under $\mathcal{S}$/$\mathcal{T}$ protocol outperforms that trained under $\mathcal{S}$/$\mathcal{Q}$ protocol.
>
> | Method | 5-way 1-shot | 5-way 5-shot |
> | :---- | :----: | :----: |
> | ProtoNet ($\mathcal{S}$/$\mathcal{Q}$ protocol) | 28.32 | 34.62 |
> | ProtoNet ($\mathcal{S}$/$\mathcal{T}$ protocol) | **34.20** | **39.77** |
>
> To summarize, when task distribution shift exists, meta-learning models trained under $\mathcal{S}$/$\mathcal{T}$ protocol can generalize better to some extent. We think there are two main reasons for this:
> - Under $\mathcal{S}$/$\mathcal{T}$ protocol, meta-model has a more accurate supervision signal (see Section 4.2 for details). Model matching helps us get rid off query sampling issues and possibly existing label noises.
> - The distillation term plays the role of a kind of regularization (see Section 5.3 for details). This prevents meta-model from over-fitting seen domain.
>
> ## Different label space
> The reviewer questioned 'if the label space of new tasks is different from the training one, will the proposed approach still be effective'. Actually, in our paper, the label space of meta-testing tasks **is different** from that of meta-training tasks. In few-shot learning experiment, meta-test tasks are composed of **non-overlapping classes** with those in meta-training stage. Thus, we have verified that our $\mathcal{S}$/$\mathcal{T}$ protocol can be applied across different label spaces.
>
> Another important question is 'what if label space of testing tasks is different from that of target models'. Please note that **we only need target models in meta-training phase** to enrich the supervision of the meta-model, so the meta-model becomes more generalizable to novel tasks. During the meta-testing stage, we do not need target models, and the learned meta-model can perform well.

---

> > ### Comment · Reviewer_mVfo · 2021-08-19
> > **Response to changes**
> >
> > Thank you. You have solved all my concerns.

---

> > > ### Author Response · Authors · 2021-08-19
> > > **Response from authors**
> > >
> > > Thank you for your response! Since our rebuttal solved all your concerns, we would appreciate it if you could change your rating and accept our paper.

---

> ### Author Response · Authors · 2021-08-10
> **Official Comment in Response to Reviewer mVfo (Part II)**
>
> [*This is the second part of our responses.*]
>
> ## Task hardness and the pre-trained model
> As the reviewer mentioned, 'the recognition of task hardness depends on the base model $\phi^{\text{pt}}$, which is pre-trained on training task dataset', and this is true. The reviewer required us to investigate the influence of the pre-trained model on final performance, and we consider it as an important complementary part of our experiment. Thank you for your constructive advice!
>
> In order to study the influence of pre-trained model, we try to change the 'strength' of the pre-trained model and test the final performance. Specifically, we obtain pre-trained models with different strengths from two aspects:
> - Data dimension: we change the number of instances per class to pre-train the model.
> - Time dimension: we save several snapshots of the pre-trained model during training process.
>
> Other experiment settings are same as those in Section 5.4. That is, we firstly use the pre-trained models to select **10% hardest tasks**, and then build target models for these tasks. We use the previous pre-trained model as model initialization, but the selected tasks are changed.
>
> Firstly, we study the influence of training data. We vary the ratio of training data in {10%, 50%, 70%, 100%}. The pre-trained model is stronger when using more training data. Final mean accuracies (5-way 1-shot and 5-way 5-shot) are shown in the following table ($\star$ means same results as those in Table 5 and Table 6). We can see that when using 10% training data, the accuracies are similar to those using random task selection (66.72 for 1-shot and 81.05 for 5-shot). This means the pre-trained model using 10% data hardly choose meaningful tasks. However, when using 50% or 70% training data, the final accuracies are improves a lot. This means the pre-trained model can to some extent select the most difficult tasks, and target models on these tasks can improve final performance.
>
> | Training Data Ratio | 10% | 50% | 70% | 100% |
> | :---- | :----: | :----: | :----: | :----: |
> | ProtoNet($\mathcal{S}$/$\mathcal{T}$) Mean Accuracy, 5-way 1-shot | 66.74 | 67.35 | 67.81 | 68.03 ($\star$) |
> | ProtoNet($\mathcal{S}$/$\mathcal{T}$) Mean Accuracy, 5-way 5-shot | 81.13 | 81.92 | 82.40 | 82.53 ($\star$) |
>
> Secondly, we study the influence of training epochs. We save four snapshots during the training process of the pre-trained model. Note that we use the whole training dataset here. Other settings are same as those in previous experiment. Results are shown in the following table ($\star$ means same results as those in Table 5 and Table 6). When we use a random initialized model (training epoch = 0) to select tasks, the final performance is not good. But when we pre-train the model for 100 or 150 epochs, it can successfully select hard tasks and improve final accuracy.
>
> | Training Epochs | 0 | 50 | 100 | 150 | 200 |
> | :---- | :----: | :----: | :----: | :----: | :----: |
> | ProtoNet($\mathcal{S}$/$\mathcal{T}$) Mean Accuracy, 5-way 1-shot | 65.92 | 66.68 | 67.15 | 67.10 | 68.03 ($\star$) |
> | ProtoNet($\mathcal{S}$/$\mathcal{T}$) Mean Accuracy, 5-way 5-shot | 80.72 | 81.05 | 81.49 | 81.77 | 82.53 ($\star$) |
>
> ## Why does our method generalize better
> The reviewer asked why $\mathcal{S}$/$\mathcal{T}$ protocol can help meta-learning algorithms generalize better, and now we will provide some explanations from two aspsects:
> - $\mathcal{S}$/$\mathcal{T}$ protocol offers a better supervision signal than randomly sampled query sets.
> - $\mathcal{S}$/$\mathcal{T}$ protocol regularizes meta-model through knowledge distillation.
>
> ### It is better to measure the quality of sovlers using target models than query sets.
> Recall that a typical meta-learning algorithm can be decomposed into two iterative phases. In the first phase, we train a solver of a task on its support set with assistance of meta-model. In the second phase, we optimize the solver's performance to update meta-model. Here a key factor is the way to evaluate the solver.
>
> Early meta-learning algorithms[12,13] directly evaluate the solver by its support loss, which may cause inner-task over-fitting. It is obviously better to evaluate the solver on **unseen** instances, and this is why $\mathcal{S}$/$\mathcal{Q}$ protocol succeeds. A recent paper[11] theoretically analyzed why meta-learning algorithms trained under $\mathcal{S}$/$\mathcal{Q}$ protocol outperform early ones. In [11], the authors concluded that the generalization bound of a meta-learning algorithm have two parts: **outer-task gap** and **inner-task gap**. Intuitively, outer-task gap can be decreased by increasing the number of training tasks, and inner-task gap can be decreased by increasing the number of query instances in each task. That's why we claim **meta-learning algorithms hurt when query sets are small**. However, if we evaluate solvers using target models, we do not need to randomly sample query instances, and inner-task gap can be closed. This is a key insight of our work, although we leave detailed theoretical analysis as future work.
>
> Moreover, sometimes query sets can be biased due to sampling issues. In Section 5.3 and Table 4, we construct a Gaussian classification dataset, and show that biased sampling of query instances can dramatically hurt meta-learning algorithms. In this case, evaluating task-specific solvers using target models help us get rid off sampling issues.
>
> In summary, $\mathcal{S}$/$\mathcal{T}$ protocol provides us with a more reliable method to evaluate solvers, so give meta-model a more accuract supervision signal.
>
> ### Target models have the effect of regularization.
> In our proposed method, we match the task-specific solver and corresponding target model in output space, which is similar to knowledge distillation. As we know, knowledge distillation has the effect to regularize model, which is similar to label smoothing[14]. Thus, meta-model trained under $\mathcal{S}$/$\mathcal{T}$ protocol are less likely to be over-fitted, and the decision boundaries are smoother (see Section 5.3 and Figure 6 for details).
>
> ## References
> [1] Finn C, Abbeel P, Levine S. Model-agnostic meta-learning for fast adaptation of deep networks[C]//International Conference on Machine Learning. PMLR, 2017: 1126-1135.
>
> [2] Snell J, Swersky K, Zemel R S. Prototypical networks for few-shot learning[J]. arXiv preprint arXiv:1703.05175, 2017.
>
> [3] Vinyals O, Blundell C, Lillicrap T, et al. Matching networks for one shot learning[J]. Advances in neural information processing systems, 2016, 29: 3630-3638.
>
> [4] Wah C, Branson S, Welinder P, et al. The caltech-ucsd birds-200-2011 dataset[J]. 2011.
>
> [5] Koh P W, Sagawa S, Xie S M, et al. Wilds: A benchmark of in-the-wild distribution shifts[C]//International Conference on Machine Learning. PMLR, 2021: 5637-5664.
>
> [6] Tseng H Y, Lee H Y, Huang J B, et al. Cross-domain few-shot classification via learned feature-wise transformation[J]. arXiv preprint arXiv:2001.08735, 2020.
>
> [7] Guo Y, Codella N C, Karlinsky L, et al. A broader study of cross-domain few-shot learning[C]//European Conference on Computer Vision. Springer, Cham, 2020: 124-141.
>
> [8] Taylor J, Earnshaw B, Mabey B, et al. RxRx1: An Image Set for Cellular Morphological Variation Across Many Experimental Batches[C]//The 7th International Conference on Learning Representations. 2019.
>
> [9] Ye H J, Hu H, Zhan D C, et al. Few-shot learning via embedding adaptation with set-to-set functions[C]//Proceedings of the IEEE/CVF Conference on Computer Vision and Pattern Recognition. 2020: 8808-8817.
>
> [10] Tripuraneni N, Jin C, Jordan M. Provable meta-learning of linear representations[C]//International Conference on Machine Learning. PMLR, 2021: 10434-10443.
>
> [11] Chen J, Wu X M, Li Y, et al. A closer look at the training strategy for modern meta-learning[J]. Advances in Neural Information Processing Systems, 2020, 33.
>
> [12] Santoro A, Bartunov S, Botvinick M, et al. Meta-learning with memory-augmented neural networks[C]//International conference on machine learning. PMLR, 2016: 1842-1850.
>
> [13] Vilalta R, Drissi Y. A perspective view and survey of meta-learning[J]. Artificial intelligence review, 2002, 18(2): 77-95.
>
> [14] Yuan L, Tay F E H, Li G, et al. Revisiting knowledge distillation via label smoothing regularization[C]//Proceedings of the IEEE/CVF Conference on Computer Vision and Pattern Recognition. 2020: 3903-3911.

---

### Official Review · Reviewer_V6R5 · 2021-07-08

**Rating:** 6
**Confidence:** 4

**Summary:**

This paper investigated the S/T meta-learning framework which is not widely used due to the huge computational cost. The proposed method estimates the hardness of tasks and selects the target models of a small ratio of hard tasks. Then, the meta-model is learned with the S/Q meta-loss and the knowledge distillation loss between the selected target models and solvers. Experiments show that the proposed method outperforms the two classic meta-learning methods with the cost of constructing target models.

**Limitations And Societal Impact:**

It would be better to explicitly discuss the limitation in the main pages. Societal impact is N/A.

**Main Review:**

1.	Generally speaking, this paper is easy to read and understand. In the introduction, it will be clearer to match the solver, target model and meta-model to the corresponding modules or neural networks. Maybe the authors can give brief idea of how to construct the target models in the introduction. The description about S/Q and S/T redundantly appears several times in the paper and can be shorten.
2.	The equivalence between the S/T meta-loss and the simplified one (Line 179-180) is interesting. Does it mean that adding random noise to training data is the key to meta-learning? Do the two losses lead to the same results? Can the authors empirically verify it?
3.	The study and analysis on the target models are sufficient and solid.
4.	The experiments on miniImageNet and tieredImageNet show that the proposed method achieves obvious improvements over the two classic meta-learning methods with the cost of constructing target models.
5.	Overall, I think this paper can to some extent advance the meta-learning research.


**Time Spent Reviewing:**

3

---

> ### Author Response · Authors · 2021-08-09
> **Official Comment in Response to Reviewer V6R5**
>
> Dear reviewer,
>
> thank you for your review and comments! We appreciate it that you consider our work as ‘easy to read and understand’ and our experiments as ‘sufficient and solid’. Now we give responses to your comments and questions.
>
> ## Writing of introduction
> The reviewer mentioned that ‘it will be clearer to match the solver, target model, and meta-model to the corresponding modules or neural networks’, and we side with you. In order to make this clearer, we will make two changes:
> - We will add a sketch figure to explain what exactly are solvers, target models, and meta-model in Section 1.
> - We will add a concrete example to describe what are solvers, target models, and meta-model. Taking meta-learning algorithm MAML as an instance, meta-model is the common model initialization shared across tasks, solvers are task-specific models trained on each task's support set, and target models are theoretically optimal (or good enough) solvers of each task.
>
> We will also describe the way to construct target models in the introduction to help readers understand what’s going on.
>
> In the paper, we describe $\mathcal{S}$/$\mathcal{Q}$ protocol and $\mathcal{S}$/$\mathcal{T}$ protocol several times because these two concepts are important. We agree to your idea that this should be shorten, and we will make the description simplified in the final version.
>
> Thank you very much for your suggestions on paper writing!
>
> ## Equivalence between $\mathcal{S}$/$\mathcal{T}$ meta-loss and the simplified one
> We will answer your questions and explain this experiment again in detail. In this experiment, we generate a synthetic regression dataset to check whether target models can improve the generalization ability of meta-model. In the data generation process, we add random noise $\epsilon$ to label $\mathcal{T}(x)$ to mimic real-world setting, i.e., $y=\mathcal{T}(x)+\epsilon$. After that, we train the meta-model based on tasks containing these noisy labels $y$.
>
> ### Q1: Does it mean that adding random noise to training data is the key to meta-learning?
> Not exactly. In this experiment, the purposes of adding random noises are two-fold. Firstly, adding random noises mimics real-world setting. Secondly, we can check how target models help to defense these random noises. That is to say, adding random noises in this experiment is **not** for improving meta-learning. In the following, we check how random noises affect meta-learning following your suggestions.
>
> Firstly, we show that random noises may hurt vanilla meta-learning. Since the noises will overwhelm the true property of data, learning with noisy labels is more difficult than learning with clean ones. To show this, we additionally construct two datasets following the generation process described in Section 4.2. We can control the ‘noisy extent’ by ranging the variance parameter in Gaussian distribution. By setting variance to a larger value, the ‘noisy extent’ is enlarged. The following table shows the average MSE losses of MAML[1] and ProtoNet[2] on three sinusoid datasets with noise variance 0.5, 1, and 5 ($\star$ means the same configuration as Table 1 in the paper). We can see that both algorithms suffer from a catastrophic performance drop on noisier datasets.
>
> | Method | MAML | ProtoNet |
> | :------ | :------: | :------: |
> | noise var = 0.5 ($\star$) | 4.933 | 4.706 |
> | noise var = 1 | 5.792 | 5.428 |
> | noise var = 5 | 9.241 | 8.921 |
>
> Secondly, we prove that $\mathcal{S}$/$\mathcal{T}$ protocol helps defense random noises. As is presented in Line 179-180, target models play the role of 'denoiser' in this experiment. We can prove that training loss under $\mathcal{S}$/$\mathcal{T}$ protocol is equivalent to the standard MSE loss with 'cleaner' labels. Here cleaner labels mean labels with 'smaller' noises, and the 'extent of cleaning' is controlled by hyper-parameter $\lambda$. Here $\lambda$ is set to 0.5. Now, we empirically verify the effect of target models on three sinusoid datasets. Results are shown in the following table ($\star$ means the same configuration as Table 1 in the paper).
>
> | Method | MAML ($\mathcal{S}$/$\mathcal{Q}$) | MAML ($\mathcal{S}$/$\mathcal{T}$) | ProtoNet ($\mathcal{S}$/$\mathcal{Q}$) | ProtoNet ($\mathcal{S}$/$\mathcal{T}$) |
> | :------ | :------: | :------: | :------: | :------: |
> | noise var = 0.5 ($\star$) | 4.933 | **3.621** | 4.706 | **3.332** |
> | noise var = 1 | 5.792 | **3.928** | 5.428 | **3.625** |
> | noise var = 5 | 9.241 | **5.292** | 8.921 | **5.013** |
>
> Two algorithms with $\mathcal{S}$/$\mathcal{T}$ protocol outperform those trained under $\mathcal{S}$/$\mathcal{Q}$ protocol because of the denoising effect. When noise variance is large (noise var=5), MAML and ProtoNet trained under $\mathcal{S}$/$\mathcal{Q}$ protocol are almost disabled, and incorporating the supervision of target models helps a lot.
>
> Moreover, **we can achieve this improvement with only a small number of target models**. Refer to Section 4.2 and Figure 4 for more details.
>
> ### Q2: Do the two losses lead to the same results? Can the authors empirically verify it?
> Since these two losses are theoretically equivalent (refer to Supp for detailed proof), they lead to same results except for negligible numerical errors.
>
> Empirically, we can train two meta-models using these two losses on the orginal sinusoid dataset with different $\lambda$ values. Results are shown in the following two tables. We can see they achieve very similar results except for numerical errors.
>
> | $\lambda$ | 1 | 0.8 | 0.5 | 0.2 |
> | :---- | :----: | :----: | :----: | :----: |
> | MAML - $\mathcal{S}$/$\mathcal{T}$ Loss | 3.220 | 3.419 | 3.621 | 3.833 |
> | MAML - Standard MSE with CLeaner Labels | 3.224 | 3.415 | 3.614 | 3.828 |
>
> | $\lambda$ | 1 | 0.8 | 0.5 | 0.2 |
> | :---- | :----: | :----: | :----: | :----: |
> | ProtoNet - $\mathcal{S}$/$\mathcal{T}$ Loss | 3.137 | 3.304| 3.332 | 3.550 |
> | ProtoNet - Standard MSE with CLeaner Labels | 3.141 | 3.301 | 3.327 | 3.555 |
>
> At last, we want to thank you for your review and kind comments like 'the study and analysis on the target models are sufficient and solid', 'the experiments on miniImageNet and tieredImageNet show that the proposed method achieves obvious improvements', and 'this paper can to some extent advance meta-learning research'.
>
> ## References
> [1] Finn C, Abbeel P, Levine S. Model-agnostic meta-learning for fast adaptation of deep networks[C]//International Conference on Machine Learning. PMLR, 2017: 1126-1135.
>
> [2] Snell J, Swersky K, Zemel R S. Prototypical networks for few-shot learning[J]. arXiv preprint arXiv:1703.05175, 2017.

---

> > ### Comment · Reviewer_V6R5 · 2021-08-17
> > **Good response!**
> >
> > Thank you for the detailed response! It clearly answered my questions and proved your statement with results.

---

> > > ### Author Response · Authors · 2021-08-19
> > > **Response from authors**
> > >
> > > Thank you for your response!

---

### Official Review · Reviewer_f3eh · 2021-07-15

**Rating:** 7
**Confidence:** 5

**Summary:**

The authors propose a novel algorithm for meta-learning, in which instead of just optimizing for the query loss, they also optimize the inner loop produced model to do as well as some larger ‘teacher’ model that is an actual expert on the task at hand. The data used to train those experts are the same data we normally use during the meta-training phase in a particular benchmark, therefore their process does not require more data, but its efficiency can either be better or worse than a model trained with the standard S/Q loss depending on how many ‘expert’ models are pretrained.


**Limitations And Societal Impact:**

The impact is the same as any other few-shot learning work undetaken in the context of deep learning. Such concerns are better described in full papers in the ethics and deep learning subfield.

**Main Review:**

Abstract:

Overall a very good abstract. However, you never defined the term ‘Target Model’ and it is one of the key terms necessary to understand what is going on. I’d recommend an explicit definition somewhere in the abstract, to ensure the reader understands.

Writing Quality:

Overall the paper reads very well, with the occasional typo, but overall it is precise, concise and very clear.

A few points I’d like to mention is, firstly, Figure 1 is not readable without zooming, and the text colour matches the surrounding box which makes it even harder to read. I recommend making the figure a bit larger, and the text kept dark.

Table 5 states the word Black in what appears to be bold, therefore confusing the reader, with the actual bold results being the best results. I recommend adapting the colours/formatting here to make it more intuitive.

Conclusion appears to be weirdly formatted for some reason. I assume the figure has some role to play there -- I recommend reformatting this bit to make it cleaner.

Novelty:

Very novel, and surprising work. Using the same data, but by producing such expert models and trying to basically distill their capabilities into the resulting adapted models in each task seems to be very powerful, generalization wise. This could be the beginning of a good meta learning penalty term! :)

I find the results very convincing as to the claims made, and I appreciate that multiple independent runs were done for each data point in the tables.

Overall quite good, assuming the abstract issue is fixed, I recommend acceptance.


**Time Spent Reviewing:**

1

---

> ### Author Response · Authors · 2021-08-09
> **Official Comment in Response to Reviewer f3eh**
>
> Dear reviewer,
>
> We have read your review carefully, and are very grateful to your comments and suggestions. We are happy to see that our work is considered as ‘precise, concise, and very clear’, and ‘novel and surprising’. Here are our responses.
>
> ## The defition of target model
> The reviewer mentioned that ‘you never defined the term Target Model and it is one of the key terms necessary to understand what is going on’, and we side with you. Introducing the concept of target model in abstract can help readers understand our work. We have modified our abstract to make the concept of target model clearer without changing abstract main body. New abstract is as follows, and we empahsize the main changes in bold. Thank you for your advice!
>
> New Abstract:
> Meta-learning can extract an inductive bias from previous learning experience and assist the training processes of new tasks. It is often realized through optimizing a meta-model with the evaluation loss of a series of task-specific solvers. Most existing algorithms sample non-overlapping support sets and query sets to train and evaluate the solvers respectively due to simplicity ($\mathcal{S}$/$\mathcal{Q}$ protocol). **Different from $\mathcal{S}$/$\mathcal{Q}$ protocol, we can also evaluate a task-specific solver by comparing it to a target model $\mathcal{T}$, which is the optimal model for this task or a model that behaves well enough on this task ($\mathcal{S}$/$\mathcal{T}$ protocol). Although being short of research, $\mathcal{S}$/$\mathcal{T}$ protocol has unique advantages such as offering more informative supervision, but it is computationally expensive**. This paper looks into this special evaluation method and takes a step towards putting it into practice. We find that with a small ratio of tasks armed with target models, classic meta-learning algorithms can be improved a lot without consuming many resources. Furthermore, we empirically verify the effectiveness of $\mathcal{S}$/$\mathcal{T}$ protocol in a typical application of meta-learning, i.e., few-shot learning. In detail, after constructing target models by fine-tuning the pre-trained network on those hard tasks, we match the task-specific solvers to target models via knowledge distillation. Experiments demonstrate the superiority of our proposal.
>
> ## Writing quality
> - We have checked our paper carefully, and will fix all typos in the final version.
> - You mentioned that ‘Figure 1 is not readable without zooming’. We will increase the font size in Figure 1, and change the text color to make it clear.
> - In Table 5, we will change the color ‘black’ to ‘blue’.
> - About the conclusion format, you are right that it is influenced by the figure. After adjusting figure format and some texts, the conclusion part will be well formatted in the final version.
>
> Finally, thank you very much for your detailed comments and useful suggestions!

---

> > ### Comment · Reviewer_f3eh · 2021-08-18
> > **Response to changes**
> >
> > Indeed I find the updated abstract a lot more clear now. Assuming you also take care of the rest of the suggestions, this should make a good entry to the NeurIPS 2021 conference. :)
> >
> > Thank you for your kind words, as well as your time and contributions.

---

> > > ### Author Response · Authors · 2021-08-19
> > > **Response from authors**
> > >
> > > Thank you for your response!

---

### Decision · Program_Chairs · 2021-09-27

**Decision:**

Accept (Poster)

**Comment:**

The reviewers agree that the paper has interesting ideas, and represents an exciting direction in meta learning domain. Hopefully, reviewer comments should help preparing the next draft.